# PromptCCD: Learning Gaussian Mixture Prompt Pool for Continual Category Discovery

## Abstract

In this paper, we address the challenging open-world learning problem of continual category discovery (CCD). Initially, a labelled dataset consisting of known categories is provided to the model. Subsequently, unlabelled data arrives continuously at different time steps, which may contain objects from known or novel categories. The primary objective of CCD is to automatically assign labels to unlabelled objects, regardless of whether they belong to seen or unseen categories. However, the crucial challenge in continual category discovery is to automatically discover new categories in the unlabelled stream without experiencing catastrophic forgetting, which remains an open problem even in conventional, fully supervised continual learning. To address this challenge, we propose PromptCCD, a simple yet effective approach that utilizes Gaussian mixture model as a prompting method for CCD. At the core of PromptCCD is our proposed Gaussian Mixture Prompt Module (GMP), which acts as a dynamic pool updating over time to provide guidance for embedding data representation and avoid forgetting during continual category discovery. Additionally, our GMP provides the unique advantage of enabling on-the-fly estimation of category numbers, which enables it to discover categories in the unlabelled stream without prior knowledge of category numbers. Finally, we extend the standard evaluation metric for generalized category discovery to CCD and benchmark state-of-the-art methods using different datasets. Our PromptCCD significantly outperforms other methods, demonstrating the effectiveness of our approach.

## 1 Introduction

The human visual system has the remarkable ability to learn and reason about novel concepts over time. For instance, humans can learn about newly discovered animals and extinct ones in different timelines with ease. This ability also extends to other concept axes such as arts, products, and more. Hence, the challenge of discovering novel visual concepts within unlabelled images over a period while retaining previously seen visual concepts becomes a critical aspect in the design of artificial visual systems. Continual category discovery (CCD) (Zhang et al., 2022) aims to empower artificial visual systems with this ability by extending the challenging open-world learning problems of novel category discovery (NCD) (Han et al., 2019) and generalized category discovery (GCD) (Vaze et al., 2022) to a continual learning scenario (see Fig. 1). By enabling artificial visual systems to learn and reason about novel concepts over time, CCD represents an essential step towards developing more intelligent and adaptive visual systems that can operate effectively in dynamic environments.

Advancements in vision foundation models have shown promise in various computer vision tasks, from image classification and object detection to more complex tasks like scene understanding (Caron et al., 2021; Oquab et al., 2023). State-of-the-art models like transformers have demonstrated their strong performance in static environments where they are trained on a fixed set of categories. Given the progress and capabilities of these foundation models, we are interested in investigating how these models can be repurposed to continually adapt to dynamic environments where they must discover and learn from new visual data categories over time.

There are two major challenges in CCD. The first challenge is *catastrophic forgetting*, a well-known issue in continual learning settings (De Lange et al., 2021). Traditional techniques for mitigating

Figure 1: Overview of the Continuous Category Discovery task. In the initial stage, the model learns from labelled data, while in the subsequent stages, the model learns from an unlabelled data stream containing instances from known and novel classes.

forgetting, such as rehearsal-based (Rebuffi et al., 2017), distillation-based (Li & Hoiem, 2017), architecture-based (Li et al., 2019), and prompting-based methods (Wang et al., 2022b;a), assume fully labelled data at each stage, which is incompatible with the CCD framework where the goal is to work with unlabelled data streams. The second challenge is the discovery of novel visual concepts. While Generalized Category Discovery (GCD) is a related task, most existing methods focus mainly on static unlabelled data, making them unsuitable for the continually evolving nature of CCD.

To tackle these challenges in adapting foundational vision models for CCD, we introduce a Gaussian Mixture Prompt learning framework. This framework employs a Gaussian Mixture Model (GMM) to model the data distribution at each learning stage dynamically. By enriching the visual feature representation with adaptive queried Gaussian Mixture Prompts (GMP), our method excels at identifying new visual concepts across successive learning stages. Concurrently, these prompts facilitate the model's seamless adaptation to emerging data while preserving its performance on previously acquired categories, thus preventing catastrophic forgetting. In addition to outperforming existing CCD solutions, our framework provides the unique advantage of enabling on-the-fly estimation of category numbers — often assumed to be predetermined in prior works (Zhang et al., 2022).

We summarize our main contributions as follows: (1) We introduce *Gaussian Mixture Prompt Module* (GMP), a new prompt learning technique that leverages Gaussian mixture component(s) to generate better representation and mitigate the catastrophic forgetting problem on previously learned data. (2) We propose the first prompt learning framework tailored for CCD, *PromptCCD*, which can be coupled with our proposed GMP and existing prompt learning techniques for effective continual category discovery. (3) We extensively experiment with benchmarking datasets and compared our method with baseline methods under both known and unknown category number scenarios, significantly outperforming the state-of-the-art.

## 2 METHOD

In GCD, given a labelled and unlabelled set of images, the task is to recognize and discover all known and novel classes in the unlabelled set. The CCD task extends this task into the continual setup where the unlabelled data stream keeps coming in different time steps. Thus, the main objective of CCD task is to discover novel classes in a dynamic setting without forgetting learned knowledge from the previous streamed data, *i.e.* a decrease in the model's performance on known categories. In this section, we briefly describe how CCD task is formulated.

Given dataset $D = D^l \cup D^u$ consisting of labelled and unlabelled data respectively. $D^l = \{(x_i, y_i)\}_{i=1}^N$ contains tuples of the input $x_i \in \mathcal{X}$ and its corresponding labels $y_i \in \mathcal{Y}$. The labelled dataset of known categories will be used for the model to learn in the initial stage. In the subsequent (discovery) stages, assuming the total number of stages is $T$, the unlabelled data stream $D^u$ is divided into $T$ subsets such that $D^u = \{D_t^u\}_{t=1}^T$ where each unlabelled set at stage $t$, $D_t^u = \{D_t^{uo}, D_t^{un}\}$, consists of unlabelled instances from known and novel categories, respectively. Our goal in CCD is to train a model $\mathcal{H}_\theta : \mathcal{X} \to \mathcal{Z}$ parameterized by $\theta$ that first, learns from labelled $D^l$ and in the discovery stages, learns from unlabelled data $D_t^u$ for time steps $T$ such that $\mathcal{H}_\theta$ can be used to discover novel classes and assign class labels to all unlabelled instances utilizing representative feature $z_i \in \mathcal{Z}$ without forgetting previously learned knowledge from old streamed data.

In the following, we first elaborate on the design of our baseline and proposed methods, followed by an explanation of how our model learns during the initial and continual discovery stages.

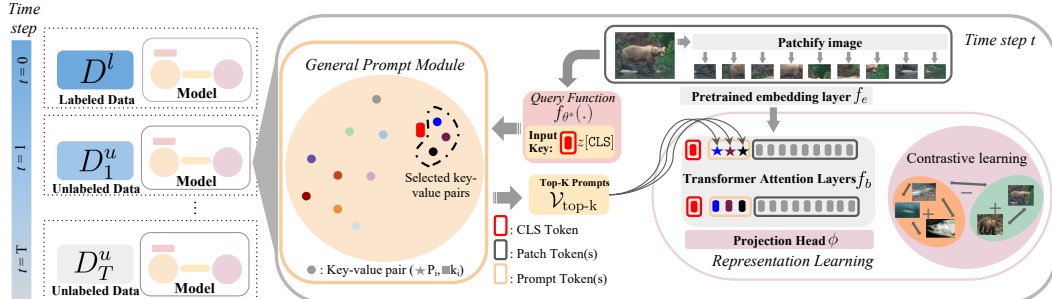

Figure 2: Our baseline prompt-based CCD framework adopts a prompt-based continual learning model that uses a prompt pool module to guide the self-supervised vision foundation model for CCD.

## 2.1 PROMPT POOL LEARNING FOR CONTINUAL CATEGORY DISCOVERY

Vision foundation models are pretrained representations trained on a large-scale dataset and are task-agnostic. These models can achieve remarkable performance across certain downstream tasks even with minimal fine-tuning. Prompt tuning (Wang et al., 2022a;b) has emerged as a powerful method for adapting these foundation models to supervised continual learning settings. However, directly utilizing these prompt learning techniques is unsuitable for CCD task, as all these works assume the incoming data stream has label information.

We start our exploration by constructing a prompt learning baseline for CCD. Inspired by Wang et al. (2022b;a), we design a baseline model for CCD that leverages a shared memory pool of prompts. The model extracts a feature from a query example using a frozen pretrained model, and the feature will be used to retrieve the top-k most relevant prompts from the fixed-size $M$ prompts in the shared pool. These prompts are then used to guide the model's representation learning by prepending them with the input's embeddings, optimised with contrastive learning at each learning stage. The formulation of the method is presented below (as depicted in Fig. 2).

Given a model $\mathcal{H}_\theta : \{\phi, f_\theta\}$. $\phi$ is a MLP projection head, and $f_\theta = \{f_e, f_b\}$ is the transformer-based feature backbone which consists of input embedding layer $f_e$ and self-attention blocks $f_b$. An input image $x \in \mathbb{R}^{H \times W \times 3}$ where $H, W$ represent the height and width of the image, is first split into L tokens (patches) such that $x_q \in \mathbb{R}^{L \times (h \times w \times 3)}$ where $h, w$ represent the height and width of the image patches. These patches are then projected to the input embedding layer such that $x_e = f_e(x_q) \in \mathbb{R}^{L \times z}$. To construct the prompt learning technique, a learnable prompt pool is initialized as $\mathbb{V} = \{V_m\}_{m=1}^M$ where $V_m \in \mathbb{R}^{L \times z}$ and $M$ is the total number of prompts (which is fixed, across stages). Additionally, a query function $f_{\theta^*} : \mathbb{R}^{H \times W \times 3} \to \mathbb{R}^{z[\text{CLS}]}$ is initialized to map $x \to$ classification token. To form the key-value memory query function, these prompts $V_i$ are then paired with learnable key $k_i$ so that, given query $f_{\theta^*}(x)$ and $\{k_m\}_{m=1}^M$ set of keys, we calculate their similarity using cosine distance $\gamma$ and take the top-k keys. With these selected keys, we can return the set of associated prompts called $\mathcal{V}_{\text{top-k}}$. Then, a set of embeddings $x_{total} = [\mathcal{V}_{\text{top-k}}; x_e]$ is formed by prepending the selected prompts with the patch embeddings. Finally, we feed the embeddings to the self-attention blocks $f_b(x_{total})$. As our baseline adopts the contrastive learning strategy, let $\{x_i, x_i'\}$ be the two views of randomly augmented image $x_i$. These two pairs are then fed to $\mathcal{H}_\theta$ such that $z_i, z_i' = \phi(f_\theta(x_i, x_i'))$. We optimize our baseline by combining contrastive learning losses, Sec. 2.3 and the surrogate loss, Eq. (1) to pull selected keys closer to corresponding query features. Finally, when stage $t$ training is finished, we transfer current prompt pool $\mathbb{V}$ to the next stage.

$$\mathcal{L}_{surrogate}(x_i, k_m) = \gamma(f_{\theta^*}(x_i), k_m). \tag{1}$$

Although this prompt learning technique by design works in our baseline, several limitations arise when applied to CCD task. First, The representation learning process only considers the unlabelled data in the current time step, which can cause representation bias towards current data and disrupt the representation learned for the previous data. Additionally, the category discovery process is separate from the representation learning process, which means that there is no proper mechanism for transferring knowledge from old classes to new classes. This transfer of knowledge is essential for the category discovery task. Second, the fixed-size prompt module can lead to parameter inefficiency and restrict the model's ability to discover new categories and avoid forgetting.

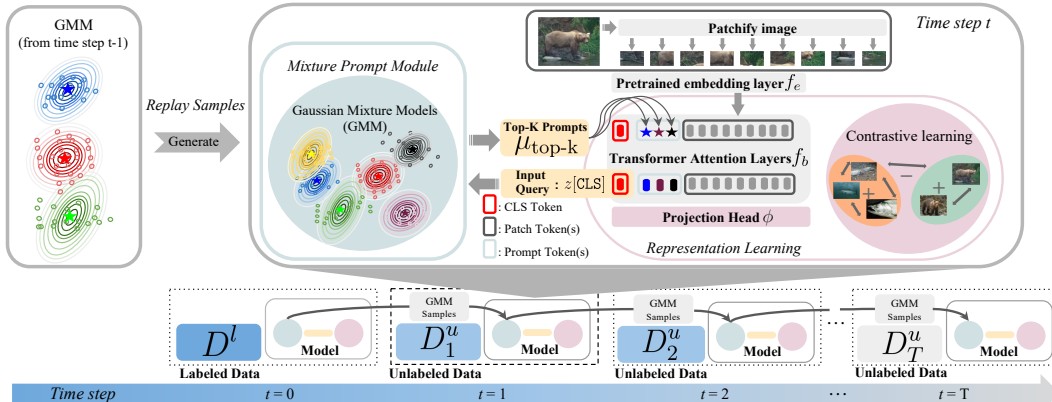

Figure 3: The design paradigm of our PromptCCD framework. PromptCCD continually discovers new categories while retaining previously discovered ones by learning a dynamic Gaussian mixture prompt (GMP) pool to guide the self-supervised vision foundation model for CCD. To prevent catastrophic forgetting, we generate replay samples from the previously fitted GMM at time step $t-1$ and fit them into the current GMM at time step $t$.

## 2.2 Gaussian mixture prompt pool learning for continual category discovery

By referring to the aforementioned limitations, there is a need for a prompting technique that requires minimal to almost zero supervision, and design-wise, its parameter has to be dynamic and flexible. With that goal in mind, here we propose the Gaussian Mixture Prompt Module (GMP), a novel prompt learning technique that uses the Gaussian mixture model (GMM) as a prompt pool. Here, we listed several key advantages that our prompt module offers; First, GMP's prompt serves a dual role, namely (1) as a task prompt to instruct the model (like in Wang et al. (2022a;b)) and (2) as class prototypes (see Appendix I for details) to act as parametric replay sample distribution for discovered classes. The second role, which is unique and important for CCD/GCD, not only allows the model to draw unlimited replay samples to facilitate the representation tuning and class discovery in the next time step but also allows the model to transfer knowledge of previously discovered and novel categories and incorporate this information when making the decision to discover a novel category. Second, our GMP module enables easy adjustment of parameters and dynamic expansion across stages. This flexibility is particularly valuable in CCD tasks where the number of classes can change over time. Finally, GMP can be seamlessly combined with a category number estimator to tackle the open-world nature of CCD, where the number of categories within the unlabelled data stream is unknown. Next, we will show how we formulate our framework, Fig. 3 utilizing GMP, Fig. 4.

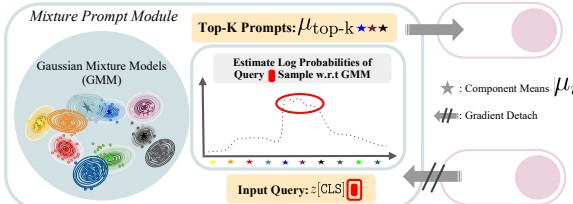

Figure 4: Our proposed Gaussian mixture prompt module (GMP) estimates the probability of the input query $z[\texttt{CLS}]$ by calculating the log-likelihood of Eq. (2). We then use the top-k mean components as prompts to guide our model for CCD.

**Gaussian mixtures prompt module (GMP).** We build a probabilistic multivariate GMM representing the presence of a sub-population within an overall population. As the Gaussian Mixture distribution is a linear superposition of Gaussian's, thus we can formulate GMM as:

$$p(z) = \sum_{c=1}^{C} \pi_c \mathcal{N}(z|\mu_c, \Sigma_c) \quad \text{s.t.} \sum_{c=1}^{C} \pi_c = 1. \tag{2}$$

Eq. (2) represents the Gaussian probability density function in GMM, consisting of $C$ Gaussian components, a set of learnable mixture weights components $\{\pi_1, \pi_2, ..., \pi_C\}$, component's mean $\{\mu_1, \mu_2, ..., \mu_C\}$, and component's covariance $\{\Sigma_1, \Sigma_2, ..., \Sigma_C\}$. When we initialize our GMM, we

assume that $C$ is known (see Sec. 2.2 for the case when $C$ is unknown). For every learning stage, we do not directly use GMM for prompting as it needs to be fit using EM algorithm by strong features, *i.e.*, $\mathcal{Z}[\texttt{CLS}] = \sum_{i=1}^{|\mathcal{X}|} f_\theta(x_i)$ , where $\mathcal{X}$ is the set of images in $D_t$. Given feature $z[\texttt{CLS}] = f_\theta(x)$ *i.e.*, the classification token queried by our model, to find the component $\pi_i$, which is associated with $z[\texttt{CLS}]$, the model calculates the log probability density value, Eq. (2) and returns a set $\mathbb{W}$ of log-likelihood probability value for different $\pi_i$. Then, we pick the top-k component's mean indexes such that top-k $= \arg\max_{\mathbb{W}' \subseteq \mathbb{W}, |\mathbb{W}'|=k} \sum_{w \in \mathbb{W}'} w$. With these selected top-k indexes, we can return the set of associated prompts as $\mu_{\text{top-k}}$ mean components. Similar to our baseline, a set of embeddings $x_{total} = [\mu_{\text{top-k}}; x_e]$ is formed by prepending the selected prompts with the patch embeddings. As our method adopts the contrastive learning strategy, let $\{x_i, x_i^{'}\}$ be the two views of randomly augmented image $x_i$. These two pairs are then fed to $\mathcal{H}_\theta$ such that $z_i, z_i^{'} = \phi(f_\theta(x_i, x_i^{'}))$. We optimize our model with contrastive learning losses, Sec. 2.3. Since we aim to use GMM dynamically across stages, once the training process is complete, we further make use of the learned GMM to draw a set of random samples $\mathcal{Z}_t^s$ by Eq. (2) where the set has $S$ samples for each component $c$. This is done to handle forgetting previously learned knowledge as the generated GMM samples $\mathcal{Z}_t^s$ from the current stage will be used to fit the next GMM. By combining samples from the previous stage and the current features *i.e.*, $\mathcal{Z}[\texttt{CLS}]_t = \sum_{i=1}^{|\mathcal{X}|} f_\theta(x) + Z_{t-1}^s$, GMM learns rich features, which leads to better prompt embeddings. The pseudo-code of the overall method is provided in Appendix A.

**Unknown number of classes in unlabelled data.** In the real open-world scenario, the number of categories $C$ is often unknown. Estimating the number of categories from unlabelled data is an important question. Existing work *i.e.*, GPC (Zhao et al., 2023) has addressed this problem using the Semi-Supervised Gaussian Mixture Model (SS-GMM) . As our prompt module is also based on GMM, we can seamlessly combine our prompt learning method with the GPC category estimator. In general, the GMM is first fit with $\mathcal{Z}[\texttt{CLS}]$ with an initial value of $C$, then an automatic *splitting-and-merging* strategy based on the Metropolis-Hastings ratio framework is used to measure the compactness and separability of clusters formed by the model. Clusters are split into two if they are separable, and two clusters are merged into one if they are cluttered. This process will continue until the optimization is finished. See Appendix D for details and experiment results.

## 2.3 OPTIMIZATION OBJECTIVES FOR DIFFERENT LEARNING STAGES

Both supervised, Eq. (3) and unsupervised, Eq. (4) contrastive losses are formulated as follows:

$$\mathcal{L}_i^s = -\frac{1}{|\mathbb{N}(i)|} \sum_{q \in \mathbb{N}(i)} \log \frac{\exp(z_i \cdot z_q / \tau)}{\sum_{j=1}^n \mathbb{1}_{[n \neq i]} \exp(z_i \cdot z_j^{'} / \tau)}, \qquad \mathcal{L}_i^u = -\log \frac{\exp(z_i \cdot z_i^{'} / \tau)}{\sum_{j=1}^n \mathbb{1}_{[n \neq i]} \exp(z_i \cdot z_j^{'} / \tau)}, \qquad (3) \qquad\qquad (4)$$

where $\mathbb{1}_{[n \neq i]}$ is an indicator so that the same image index will not be considered a negative pair, $\tau$ is the temperature value, and $\mathbb{N}(i)$ is the set of images with the same label $y$ in a mini-batch $B$.

**Optimization during initial learning from labelled data.** Given labelled data stream $D^l$ in the initial stage, the model optimizes both supervised, Eq. (3) and unsupervised, Eq. (4) contrastive losses. The total loss over the batch is formalized as Eq. (5), where $B^L$ denotes the labelled images in $B$ and $\lambda$ is the weighting coefficient.

**Optimization during class discovery from unlabelled data.** After learning feature representation in the initial stage, the model proceeds to the discovery stage, where the incoming data stream is unlabelled $D_t^u = \{D_t^{uo}, D_t^{un}\}$. Similar to the initial stage, the model adopts the self-supervised learning strategy, and we use unsupervised contrastive learning with a loss formulated in Eq. (4). Thus, the loss over the batch is formalized as Eq. (5) without the supervised contrastive loss, Eq. (3).

$$\mathcal{L}_{total} = (1 - \lambda) \sum_{i \in B} \mathcal{L}_i^u + \lambda \sum_{i \in B^L} \mathcal{L}_i^s. \qquad (5)$$

## 3 EXPERIMENTS

To assess the performance of our proposed framework compared with other models, we evaluate and compare PromptCCD framework with the state-of-the-art continual category discovery, generalized category discovery and continual learning models on generic image datasets and the more challenging fine-grained image dataset. Thus, in this section, we describe our experimental setups in Sec. 3.1;

then, we present our main experimental results in Sec. 3.2. Finally, ablation studies in Sec. 3.3 are conducted to verify our model's effectiveness.

## 3.1 EXPERIMENTAL SETUPS

**Datasets.** We conduct our experiments on various benchmark datasets, namely CIFAR-100 (C 100) (Krizhevsky & Hinton, 2009), ImageNet-100 (IN 100) (Russakovsky et al., 2015), TinyImageNet (Tiny 200) (Le & Yang, 2015), and the Caltech-UCSD Birds-200-2011 (CUB-200) (Wah et al., 2011). (1) CIFAR-100 contains 100 classes with 600 images per class. It is divided into 500 training and 100 testing images per class. (2) ImageNet100, contains 100 classes with 1350 images per class. it is divided into 1300 training and 50 testing images per class. (3) In TinyImageNet, there are 100,000 images divided into 200 classes. Each class has 500 training images, 50 validation images, and 50 test images. We use its training and test images in our experiments. (4) Lastly, CUB-200 is a fine-grained visual categorization dataset with 11,788 images of 200 bird species.

| Algorithm 1 CCD evaluation metric |
|---|
| **Input:** $f(.)$ models for each stage in $\{1, \cdots, T\}$ and datasets $\{D^L, D^U\}$. |
| **Output:** The ACC outputs for every stage. |
| 1: Initialize set $\mathbb{A}^L = \{D^l\}$. |
| 2: **for** $t \in \{1, \cdots, T\}$ **do** |
| 3:     $ACC_t$ = SS-Kmeans(Model: $f_t(.)$, Labelled set: $\mathbb{A}^L$, Unlabelled set: $\{D_t^u\}$) |
| 4:     Use labels assigned by SS-Kmeans such that $D_t^{u^*} \leftarrow D_t^u$ |
| 5:     Append $D_t^{u^*}$ to $\mathbb{A}^L$ |
| 6: **return** $\{ACC_t \mid t = 1, \ldots, T\}$ |

Table 1: Data distribution in CCD task.

| Class splits | Stage 0 | Stage 1 | Stage 2 | Stage 3 |
|---|---|---|---|---|
| $\{y_i \mid y_i < 0.7 * |\mathcal{Y}|\}$ | 87% | 7% | 3% | 3% |
| $\{y_i \mid 0.7 * |\mathcal{Y}| \leq y_i < 0.8 * |\mathcal{Y}|\}$ | 0% | 70% | 20% | 10% |
| $\{y_i \mid 0.8 * |\mathcal{Y}| \leq y_i < 0.9 * |\mathcal{Y}|\}$ | 0% | 0% | 90% | 10% |
| $\{y_i \mid 0.9 * |\mathcal{Y}| \leq y_i < |\mathcal{Y}|\}$ | 0% | 0% | 0% | 100% |

**Implementation details.** We use ViT-B/16 backbone (Dosovitskiy et al., 2021) initialized with DINO self-supervised vision foundation features (Caron et al., 2021) for all experiments. Please note that Wang et al. (2022b;a) utilized a well-pretrained model with supervision, which is suitable for the standard supervised continual learning task. However, it is not allowed to use such models for CCD task due to label information leakage. During training, only the final block of the vision transformer is finetuned 200 epochs with a batch size of 128, using an SGD optimizer and a cosine decay learning rate scheduler with an initial learning rate of 0.1 and minimum learning rate of 0.0001, and weight decay of 0.00005. For the mixture prompt module, we optimize the GMM every 30 epochs and start the prompt learning when the epoch is greater than 30. We set top-$k$ to be 5, and the number of GMM samples to 100. We pick the final model by selecting the best performing model on "old acc" using the validation set (evaluated every 10 epochs). All input images are resized to $224 \times 224$ and normalized to match the DINO pretrained model settings. For our proposed method, we follow the standard practice of self-supervised learning training procedure by training a base encoder/backbone $f_b$ and a projection head $\phi$ to maximize the agreement using a contrastive loss with $\lambda = 0.35$. For other compared methods, we chose the right hyper-parameters following their original papers. Finally, for the class number estimation, we follow the procedures proposed by GCD (Vaze et al., 2022) *i.e.*, by utilizing GCD's class number estimation method on DINO features with a binary search algorithm within the range of $[|\mathcal{Y}_L|, 1000]$ across all datasets and GPC (Zhao et al., 2023) dynamic class number estimation; We build our proposed framework with *PyTorch* library, trained in a single NVIDIA RTX 3090 GPU.

**Experiment settings and evaluation metrics.** CCD task consists of several stages. We set the number of stages to 4 with a specific CCD's split ratio presented in Table 1 following the setup of Zhang et al. (2022). The model is fine-tuned in each stage. During test time, the output classification token [CLS] features are used for clustering. For the clustering algorithm and label assignment, we use semi-supervised k-means (Vaze et al., 2022) on the training sets at stage $t$ and measure the clustering quality given the ground truth labels $y_i$ and the model's clustering prediction $\hat{y}_i$ such that: $ACC = \max_{g \in \mathcal{G}(\mathcal{Y}_U)} \frac{1}{|D_t^u|} \sum_{i=1}^{|D_t^u|} \mathbb{1}\{y_i = g(\hat{y}_i)\}$, where $\mathcal{G}(\mathcal{Y}_\mathcal{U})$ represents set of all permutations of class labels in the unlabelled set $D_t^u$. For the evaluation metrics across stages, we use the clustering accuracy ACC consisting of 'All', 'Old' and 'New' sets of metrics. 'All' indicates the overall accuracy on the entire set $D_t^U$, 'Old' and 'New' indicate the accuracy from instances of unlabelled data from $D_t^{L^*}$ and $D_t^U$ respectively. To properly measure the performance on the CCD task, we extend the commonly used ACC for static data into the CCD setting, as shown in Algorithm 1. Here, we use labelled data from $\{D^l, \Sigma_{i=1}^{t-1} D_i^{u^*}\}$ to help guide SS-Kmeans clustering algorithm. We set $u^* \leftarrow u$ to indicate that we use predicted labels on previously unlabelled data $D_i^u$ from the previous stage.

**Baselines.** We compare our method with the other representative CCD, GCD and continual learning models approaches, including 1) Grow and Merge (GM) (Zhang et al., 2022); 2) ORCA (Cao

et al., 2022); 3) GCD (Vaze et al., 2022); 4) SimGCD (Wen et al., 2023); 5) L2P (Wang et al., 2022b); and 6) DualPrompt (DP) (Wang et al., 2022a). As GM's encoder is based on ResNet-18 network (He et al., 2016), we re-implement their dynamic branch mechanism with the vision transformer backbone network and observe improved performance for their method compared to their original results; see Appendix E for details. We also re-implement GCD and SimGCD to suit the continual learning settings further. We integrate a replay-based method into these model where, for each stage, the model saves several samples for each discovered class and mix these samples with the next incoming streamed images. Lastly, we adopt L2P's and Dual Prompt's prompt pool module and their corresponding surrogate loss, for our baseline model and integrate it with our framework.

## 3.2 MAIN RESULTS

Table 2: Results on various coarse and fine-grained datasets where C is known in each unlabelled set.

| | Model | Stage 1 ACC (%) | | | Stage 2 ACC (%) | | | Stage 3 ACC (%) | | | Average ACC (%) | | |
|---|---|---|---|---|---|---|---|---|---|---|---|---|---|
| | | All | Old | New | All | Old | New | All | Old | New | All | Old | New |
| CIFAR 100 | ORCA (Cao et al., 2022) | 62.05 | 71.55 | 55.40 | 63.21 | 67.14 | 62.45 | 55.79 | 65.05 | 54.17 | 60.35 | 67.91 | 57.34 |
| | GCD (Vaze et al., 2022) | 85.11 | 88.61 | 82.66 | 72.18 | 69.33 | 72.73 | 63.59 | 63.14 | 63.67 | 73.62 | 73.69 | 73.02 |
| | SimGCD (Wen et al., 2023) | 65.33 | 89.68 | 48.29 | 54.89 | 67.36 | 52.51 | 32.21 | 52.77 | 28.61 | 50.81 | 69.94 | 43.14 |
| | GCD w/replay | 71.28 | 82.00 | 63.77 | 66.52 | 72.48 | 65.38 | 57.45 | 69.52 | 55.33 | 65.08 | 74.67 | 61.49 |
| | SimGCD w/replay | 50.97 | 75.31 | 33.94 | 42.03 | 62.19 | 38.18 | 40.48 | 57.62 | 37.48 | 44.49 | 65.04 | 36.53 |
| | Grow and Merge (Zhang et al., 2022) | 64.77 | 70.49 | 60.77 | 58.31 | 62.95 | 57.42 | 48.82 | 56.00 | 47.57 | 57.30 | 63.14 | 55.25 |
| | **PromptCCD w/L2P** | 86.77 | 79.76 | 91.69 | 85.05 | 64.10 | 89.05 | 73.45 | 56.95 | 76.33 | 81.75 | 66.94 | 85.69 |
| | **PromptCCD w/DP** | 76.55 | 82.98 | 72.06 | 65.05 | 75.33 | 63.09 | 61.08 | 73.53 | 58.90 | 67.56 | 77.26 | 64.68 |
| | **PromptCCD w/GMP (Ours)** | 90.20 | 90.73 | 92.51 | 85.83 | 75.62 | 87.78 | 76.64 | 67.14 | 78.30 | **84.22** | **77.83** | **86.20** |
| ImageNet 100 | ORCA (Cao et al., 2022) | 79.03 | 78.29 | 79.54 | 71.53 | 77.05 | 70.47 | 68.77 | 77.33 | 67.27 | 73.11 | 77.56 | 72.43 |
| | GCD (Vaze et al., 2022) | 82.45 | 83.51 | 81.71 | 82.27 | 78.57 | 82.98 | 81.39 | 79.14 | 81.78 | 82.03 | 80.40 | 82.15 |
| | SimGCD (Wen et al., 2023) | 83.70 | 84.04 | 81.29 | 70.08 | 78.29 | 41.35 | 70.92 | 76.57 | 57.73 | 74.90 | 79.63 | 60.12 |
| | GCD w/replay | 79.75 | 80.82 | 79.00 | 71.07 | 78.38 | 69.57 | 64.40 | 78.29 | 61.97 | 71.74 | 79.16 | 70.18 |
| | SimGCD w/replay | 59.78 | 80.00 | 45.63 | 49.36 | 64.10 | 46.55 | 41.35 | 58.48 | 38.35 | 50.16 | 67.53 | 43.51 |
| | Grow and Merge (Zhang et al., 2022) | 75.45 | 76.86 | 74.46 | 72.52 | 75.24 | 72.00 | 68.23 | 74.38 | 67.15 | 72.07 | 75.49 | 71.20 |
| | **PromptCCD w/L2P** | 81.95 | 80.69 | 82.83 | 65.77 | 73.81 | 64.24 | 66.52 | 73.05 | 65.38 | 71.41 | 75.85 | 70.82 |
| | **PromptCCD w/DP** | 77.87 | 84.57 | 73.17 | 70.17 | 83.43 | 67.64 | 66.38 | 84.10 | 63.28 | 71.47 | **84.03** | 68.03 |
| | **PromptCCD w/GMP (Ours)** | 84.62 | 84.29 | 84.86 | 80.06 | 79.62 | 80.15 | 82.75 | 77.62 | 83.65 | **82.47** | 80.51 | **82.88** |
| TinyImageNet | ORCA (Cao et al., 2022) | 59.98 | 66.90 | 55.14 | 53.69 | 60.52 | 52.39 | 55.51 | 55.95 | 55.43 | 56.39 | 61.12 | 54.32 |
| | GCD (Vaze et al., 2022) | 65.81 | 70.73 | 62.36 | 59.34 | 58.00 | 59.59 | 51.01 | 54.52 | 50.39 | 58.72 | 61.08 | 57.44 |
| | SimGCD (Wen et al., 2023) | 49.41 | 68.92 | 35.76 | 37.60 | 57.76 | 33.75 | 32.75 | 52.76 | 29.25 | 39.92 | 59.81 | 32.92 |
| | GCD w/replay | 63.83 | 65.98 | 62.33 | 58.03 | 58.81 | 57.88 | 55.16 | 58.48 | 54.58 | 59.01 | 61.09 | 58.26 |
| | SimGCD w/replay | 41.82 | 52.45 | 34.37 | 34.18 | 32.52 | 34.50 | 31.84 | 26.71 | 32.73 | 35.95 | 37.23 | 33.87 |
| | Grow and Merge (Zhang et al., 2022) | 57.91 | 63.24 | 54.31 | 46.80 | 54.29 | 45.37 | 49.30 | 53.57 | 48.56 | 51.34 | 57.03 | 50.41 |
| | **PromptCCD w/L2P** | 69.92 | 64.14 | 73.96 | 68.69 | 59.76 | 70.40 | 56.96 | 56.81 | 57.68 | 65.19 | 58.90 | 67.34 |
| | **PromptCCD w/DP** | 69.36 | 69.31 | 69.40 | 67.57 | 60.48 | 68.93 | 56.08 | 57.71 | 55.79 | 64.33 | 62.50 | 64.71 |
| | **PromptCCD w/GMP (Ours)** | 72.75 | 72.65 | 72.81 | 62.01 | 59.71 | 62.45 | 65.16 | 56.76 | 67.19 | **66.64** | **63.04** | **67.48** |
| CUB 200 | ORCA (Cao et al., 2022) | 49.79 | 66.43 | 38.66 | 31.50 | 65.71 | 24.24 | 43.71 | 70.00 | 38.58 | 41.67 | 67.38 | 33.83 |
| | GCD (Vaze et al., 2022) | 59.66 | 78.21 | 47.36 | 49.38 | 72.14 | 44.55 | 57.34 | 72.14 | 54.46 | 55.46 | 74.16 | 48.79 |
| | SimGCD (Wen et al., 2023) | 44.06 | 65.00 | 30.07 | 32.50 | 63.57 | 25.91 | 33.80 | 65.71 | 27.58 | 36.79 | 64.76 | 27.85 |
| | GCD w/replay | 56.71 | 82.14 | 48.21 | 48.63 | 77.14 | 42.58 | 53.81 | 67.50 | 46.42 | 53.05 | 75.59 | 46.42 |
| | SimGCD w/replay | 38.82 | 62.86 | 30.79 | 34.88 | 52.14 | 31.21 | 38.08 | 46.79 | 34.68 | 37.26 | 53.94 | 32.23 |
| | Grow and Merge (Zhang et al., 2022) | 38.64 | 70.71 | 27.92 | 29.25 | 65.71 | 21.52 | 44.29 | 56.07 | 39.69 | 37.53 | 64.16 | 29.71 |
| | **PromptCCD w/L2P** | 50.63 | 73.57 | 42.96 | 52.38 | 72.14 | 48.18 | 60.12 | 54.81 | 49.23 | 54.38 | 71.67 | 49.23 |
| | **PromptCCD w/DP** | 59.94 | 79.64 | 46.78 | 49.63 | 75.00 | 44.24 | 61.19 | 77.68 | 57.94 | 56.92 | 77.50 | 49.65 |
| | **PromptCCD w/GMP (Ours)** | 59.39 | 82.86 | 51.55 | 56.25 | 79.29 | 51.36 | 65.43 | 73.21 | 62.40 | **60.36** | **78.45** | **55.10** |

**Quantitative analysis.** We evaluate our method in two scenarios: when the class number $C$, is known (Table 2) in each unlabelled set at different stages, and when the class number is unknown (Table 3). (1) Table 2 shows the CCD evaluation results on generic and fine-grained datasets where each unlabelled set's class number, $C$, is known at different stages. Overall, PromptCCD w/GMP outperforms the other methods in all datasets across all instances ('All', 'Old', 'New') accuracy. As our base model is based on GCD (Vaze et al., 2022), we show that simply integrating our Gaussian mixture prompt module can sufficiently improve a static GCD model and can adapt in the CCD setting. We argue that not all prompting techniques effectively solve CCD task. By comparing our model with PromptCCD w/{L2P, DP} (our baselines), we observe that our model can handle class scaling better, as shown in Table 2, where our model performs better in both 'Old' and 'New' accuracy while the baselines suffer from performance loss at the later stage. We hypothesize that this performance drop is because their prompt pool parameters are not scalable, which limits the model's prompt technique to "instruct" the model when the number of parameters needed to learn or preserve is growing over time. Unlike our baseline, our prompting technique is scalable as we build our pool of prompts based on the Gaussian mixture models. To prevent forgetting, we can preserve previous knowledge by sampling each learned mixture component and using these samples to fit the next GMM. (2) To show the performance comparison for each model in a more realistic setting where $C$ is unknown, we also report the benchmark results in Table 3, where we only show three representative models i.e., GCD, Grow and Merge, and our model. Our method consistently outperforms all other methods by a large margin across the board, demonstrating the superior performance of our approach in the more realistic case when the class number is unknown.

Table 3: Results on various coarse and fine-grained datasets where C is unknown in each unlabelled set. Here, we estimate C for all methods using (Vaze et al., 2022) C-est algorithm on DINO features.

| | Model | Stage 1 ACC (%) | | | Stage 2 ACC (%) | | | Stage 3 ACC (%) | | | Average ACC (%) | | |
|---|---|---|---|---|---|---|---|---|---|---|---|---|---|
| | | All | Old | New | All | Old | New | All | Old | New | All | Old | New |
| **C 100** | Estimated $C$ | $C^{EST}$: 84, $C^{GT}$: 80 | | | $C^{EST}$: 84, $C^{GT}$: 90 | | | $C^{EST}$: 84, $C^{GT}$: 100 | | | | | |
| | GCD (Vaze et al., 2022) | 83.26 | 84.12 | 82.66 | 71.94 | 71.14 | 72.09 | 63.39 | 59.62 | 64.05 | 72.86 | 71.63 | 72.93 |
| | Grow and Merge (Zhang et al., 2022) | 63.43 | 72.29 | 57.23 | 57.56 | 57.52 | 57.56 | 54.51 | 51.05 | 55.12 | 58.50 | 60.29 | 56.64 |
| | **PromptCCD w/GMP (Ours)** | 90.13 | 90.45 | 91.60 | 78.32 | 73.81 | 79.18 | 75.89 | 64.76 | 77.83 | **81.44** | **76.34** | **82.87** |
| **IN 100** | Estimated $C$ | $C^{EST}$: 90, $C^{GT}$: 80 | | | $C^{EST}$: 90, $C^{GT}$: 90 | | | $C^{EST}$: 91, $C^{GT}$: 100 | | | | | |
| | GCD (Vaze et al., 2022) | 73.08 | 84.90 | 64.80 | 75.50 | 73.14 | 75.95 | 64.99 | 76.73 | 63.75 | 71.19 | **78.26** | 68.17 |
| | Grow and Merge (Zhang et al., 2022) | 64.61 | 76.73 | 56.11 | 51.18 | 72.10 | 47.18 | 57.13 | 72.19 | 54.50 | 57.64 | 73.67 | 52.59 |
| | **PromptCCD w/GMP (Ours)** | 78.21 | 77.62 | 78.57 | 76.40 | 72.29 | 78.00 | 69.83 | 76.67 | 68.63 | **74.81** | 75.53 | **75.07** |
| **Tiny 200** | Estimated $C$ | $C^{EST}$: 169, $C^{GT}$: 160 | | | $C^{EST}$: 169, $C^{GT}$: 180 | | | $C^{EST}$: 172, $C^{GT}$: 200 | | | | | |
| | GCD (Vaze et al., 2022) | 65.00 | 73.10 | 60.14 | 57.18 | 58.71 | 56.89 | 48.82 | 53.67 | 47.97 | 57.00 | 61.83 | 55.00 |
| | Grow and Merge (Zhang et al., 2022) | 57.77 | 63.00 | 54.11 | 41.16 | 53.57 | 38.00 | 51.00 | 50.43 | 51.10 | 49.98 | 53.43 | 47.74 |
| | **PromptCCD w/GMP (Ours)** | 66.96 | 72.86 | 63.43 | 61.96 | 59.14 | 62.50 | 58.94 | 58.14 | 59.08 | **62.62** | **63.38** | **61.67** |
| **CUB 200** | Estimated $C$ | $C^{EST}$: 166, $C^{GT}$: 160 | | | $C^{EST}$: 192, $C^{GT}$: 180 | | | $C^{EST}$: 220, $C^{GT}$: 200 | | | | | |
| | GCD (Vaze et al., 2022) | 52.51 | 68.28 | 42.16 | 45.36 | 70.07 | 40.15 | 54.20 | 70.71 | 50.97 | 50.69 | 69.69 | 44.42 |
| | Grow and Merge (Zhang et al., 2022) | 43.20 | 62.50 | 30.31 | 31.62 | 67.14 | 24.09 | 33.49 | 50.83 | 28.14 | 36.10 | 60.16 | 27.51 |
| | **PromptCCD w/GMP (Ours)** | 57.94 | 77.50 | 44.87 | 53.00 | 76.43 | 48.03 | 63.99 | 77.14 | 61.42 | **58.31** | **77.02** | **51.44** |

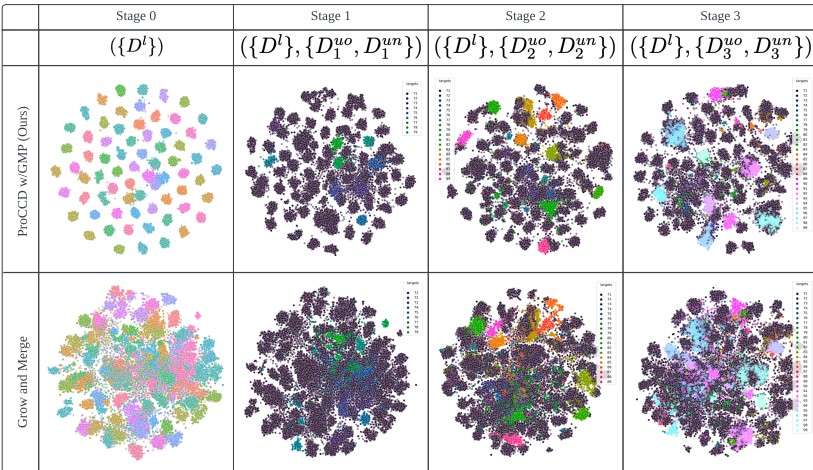

| | Stage 0 | Stage 1 | Stage 2 | Stage 3 |
|---|---|---|---|---|
| | $(\{D^l\})$ | $(\{D^l\}, \{D_1^{uo}, D_1^{un}\})$ | $(\{D^l\}, \{D_2^{uo}, D_2^{un}\})$ | $(\{D^l\}, \{D_3^{uo}, D_3^{un}\})$ |

Figure 5: TSNE visualization of CIFAR100 with features from our model PromptCCD w/GMP and Grow and Merge with DINO encoder in each stage following Table 1 distribution.

**Qualitative analysis.** Lastly, to visualize the feature representation generated by our method, we use t-SNE algorithm (Van der Maaten & Hinton, 2008) to project the high-dimensional features of $\{D^l, D_t^u\}$ in each stage into low-dimensional space. For the sake of comparison, we also provide the visualization for the feature representation generated by Grow and Merge (Zhang et al., 2022). The qualitative visualization can be seen in Fig. 5; nodes of the same colour indicate that the instances belong to the same category. Moreover, for stage t > 0, we only highlight the feature's node belonging to unknown categories. It is observed that across stages, our cluster features are more discriminative.

## 3.3 ABLATION STUDIES

Table 4: Ablation study on different components of our approach

| Covariance Type | No. Prompt | No. GMM Sampling | Sup.Contrastive | C100 Avg ACC (%) | | | CUB200 Avg ACC (%) | | |
|---|---|---|---|---|---|---|---|---|---|
| | | | | All | Old | New | All | Old | New |
| N/A | 0 prompt | 0 sample | ✓ | 73.62 | 73.69 | 73.02 | 55.46 | 74.16 | 48.79 |
| Diagonal | 5 prompts | 100 samples | ✗ | 57.86 | 65.18 | 54.59 | 33.29 | 53.87 | 26.69 |
| Diagonal | 2 prompts | 100 samples | ✓ | 79.02 | 75.21 | 80.00 | 57.24 | 77.26 | 51.50 |
| Diagonal | 5 prompts | 100 samples | ✓ | 80.69 | 76.26 | 81.48 | 59.16 | 78.21 | 53.65 |
| Diagonal | 10 prompts | 100 samples | ✓ | 80.54 | 73.92 | 83.56 | 60.28 | 77.73 | 54.06 |
| Diagonal | 5 prompts | 0 samples | ✓ | 80.33 | 72.23 | 83.17 | 57.84 | 75.05 | 51.91 |
| Diagonal | 5 prompts | 20 samples | ✓ | 80.18 | 73.89 | 80.60 | 58.87 | 76.67 | 52.44 |
| Full | 5 prompts | 100 samples | ✓ | 78.59 | 76.81 | 78.56 | **60.36** | **78.45** | **55.10** |
| Spherical | 5 prompts | 100 samples | ✓ | **84.22** | **77.83** | **86.20** | 60.06 | 75.71 | 54.01 |

To investigate the effectiveness of our Gaussian mixture-based prompt, we analyzed each component in our prompt module and present the results in Table 4. The results show a clear advantage of

adopting the Gaussian mixture prompt into our model. The number of prompts, type of covariance, and number of GMM sampling are identified as important factors. For the CIFAR-100 and CUB-200 datasets, the optimal number of prompts is five, and the number of GMM samples is 100. Regarding GMM's covariance type, "Spherical" is found to be better for CIFAR-100, while "FULL" covariance type is better for CUB-200. The default configurations are "Diagonal" for covariance type, top 5 for prompt selection, and 100 samples for GMM sampling, which appears to be a good trade-off.

## 4 RELATED WORK

**Novel / Generalized category discovery** is proposed to address the setting where there could be novel categories in the unlabelled dataset, and the goal is to automatically cluster those novel categories together (Han et al., 2019; 2021). Novel category discovery (NCD) assumes no overlap between the unlabelled and labelled data (Han et al., 2019; Zhao & Han, 2021; Fini et al., 2021), while generalized category discovery (GCD) (Vaze et al., 2022) is proposed to consider the setting where the categories in the unlabelled set can come from both the known and novel categories. It has been shown that self-supervised pretrained representations (Caron et al., 2021) greatly aid category discovery (Vaze et al., 2022). Vaze et al. (2022) further finetunes the model pretrained using one SSL contrastive loss (Chen et al., 2020) and one supervised contrastive loss (Khosla et al., 2020). Label assignment is done using a semi-supervised $k$-means algorithm. SimGCD (Wen et al., 2023) investigated the performance of parametric classifiers of different design choices, providing a strong baseline for GCD. Other works have proposed to focus on fine-grained categories (Fei et al., 2022), automatic category estimation (Hao et al., 2023; Zhao et al., 2023) , and prompt learning (Zhang et al., 2023).

**Continual learning** aims to train models that can learn to perform on a sequence of tasks, with the restriction of the model can only see the data for the current task it is trained on (De Lange et al., 2021). Catastrophic forgetting (McCloskey & Cohen, 1989) is a phenomenon that when the model is trained on a new task, it will quickly forget the knowledge on the task it has been trained on before, resulting in a catastrophic reduction of performance on the old tasks. There exists a rich literature on designing methods that enable the model to both learn to do the new task and maintain the knowledge of old tasks (Rebuffi et al., 2017; Li & Hoiem, 2017; Li et al., 2019; Wang et al., 2022b; Graves et al., 2016; Boschini et al., 2022; Buzzega et al., 2020). However, these works all assume that the incoming tasks have all the labels for the data. In contrast, in our considered setting, we assume that the new data is fully unlabelled and can have category overlap with previous tasks.

**Continual category discovery** (CCD) is a newly proposed setting with limited explorations (Zhang et al., 2022; Joseph et al., 2022; Liu et al., 2023; Roy et al., 2022). A setting termed class-iNCD is proposed by Roy et al. (2022), which is a two-stage setting where the model is first trained on a set of labelled data and then a set of only unlabelled data is provided where there is no class overlap between the two sets. Roy et al. (2022) proposed FRoST that performs replay using the feature prototypes learned on the labelled data during the discovery phase to prevent forgetting. Feature distillation and mutual information-based regularizers have also been shown to be effective for this task in NCDwF (Joseph et al., 2022). MSc-iNCD (Liu et al., 2023) extends this setting to multiple stages, and it is shown that a large pretrained model could greatly improve the performance of the discovery performance of novel categories in each of the multiple stages. Grow and Merge (Zhang et al., 2022) also tackles a similar multi-stage discovery setting, and a method containing a growing phase and a merge phase is proposed; the growing phase will use novelty detection to detect the novel categories and train the model to perform NCD; the merge phase combines the learned knowledge of the novel categories with the previous categories into a single model. A recently proposed setting termed IGCD (Zhao & Mac Aodha, 2023) considers a similar setting with MSc-iNCD, and a dataset based on the iNaturalist website is created. The most related work to ours is Zhang et al. (2022). In our paper, we adopt the data splits from Zhang et al. (2022) and propose a Gaussian mixture-based prompt learning framework to handle the task of CCD, showing superior performance.

## 5 CONCLUSION

This paper proposes a novel approach for the continual category discovery task. Our proposed model is prompt-based, utilizing Gaussian mixture components that act as an "instruction" for the model to generate better representation features. We evaluate our approach on generic image recognition and fine-grained datasets and show that it outperforms previous methods. Our experimental results demonstrate the effectiveness of our approach in the open-world setting and showcase the potential of prompt-based models for the CCD task.

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

# Appendices

# A   PSEUDO CODE FOR PROMPTCCD W/GMP

---

**Algorithm 2** PromptCCD w/GMP's Pseudo code (see Sec. 2.2 for details)

---

**Require:** $\mathcal{H}_\theta : \{\phi, f_\theta\}$ where $f_\theta : \{f_e, f_b\}$.

**Require:** GMP(.) prompt module where it contains $GMM_t$.

**Require:** Dataloader $\mathcal{B}$ for dataset $D_t$ at stage $t$.

1: Set $\alpha \leftarrow$ integer value for the incremental update epoch.
2: Set $\beta \leftarrow$ integer value for the warmup epoch.
3: **procedure** PROMPTCCD( $\mathcal{H}_\theta$, GMP(.), $\mathcal{B}$) at stage $t$
4:     /* *************************** *Model training* *************************** */
5:     **for** $e \in Epochs$ **do**
6:
7:         /* *fit $GMM_t$ every n increment of epoch*                                    */
8:         **if** $e \pmod \alpha = 0$ **then**
9:             $\mathcal{Z}[\text{CLS}]_t = \sum_{i=1}^{|\mathcal{X}|} f_\theta(x_i)$ *// extract features from $f_\theta$(.) [gradient detached]*
10:             **if** $t > 0$ **then**
11:                 $\mathcal{Z}[\text{CLS}]_t \mathrel{+}= Z_{t-1}^s$          *// combine with generated samples from $GMM_{t-1}$*
12:             OPTIMIZE($GMM_t$) by Fitting it with $\mathcal{Z}[\text{CLS}]_t$
13:
14:         **for** $B \in \mathcal{B}$ **do**          *// for simplicity, assume B batch only contains single set$\{x_i, x_i^{'}\}$*
15:
16:             /* *the next lines covered in this box describe how to acquire $\mu_{top\text{-}k}$.*          */
17:             **if** $e > \beta$ **then**                              *// when the model reaches the warm-up epoch*
18:                 $z[\text{CLS}] = f_\theta(x_i)$     *// extract features from $f_\theta$(.) [gradient detached]*
19:                 $\mu_{\text{top-k}} = \text{GMP}(z[\text{CLS}]|GMM_t)$                              *// see Fig. 4 for details*
20:             **else**
21:                 $\mu_{\text{top-k}} = None$
22:
23:             /* *the next lines covered in this box describe how $x_i$ and $\mu_{top\text{-}k}$ are projected into*
                    *the model [note: same operation for $x_i^{'}$].*                              */
24:             $x_q = \text{PATCHIFY}(x_i)$                              *// patchify image $x_i$ into L tokens*
25:             $x_e = f_e(x_q)$                              *// project to pretrained patch embedding layer*
26:             $x_{total} = [\mu_{\text{top-k}}; x_e]$                              *// concatenate $x_e$ with the $\mu_{top\text{-}k}$ prompts*
27:             $z_i = \phi(f_b(x_{total}))$                    *// project to self-attention blocks and projection head*
28:             /* *to summarize above operations, from $\mathcal{H}_\theta : \{\phi, f_\theta\}$, we got:*                    */
29:             $z_i, z_i^{'} = \phi(f_\theta(x_i, x_i^{'}))$
30:
31:         /* *optimize $\mathcal{H}_\theta$ by calculating $\mathcal{L}$ according Sec. 2.3 and do [gradient update].* */
32:         OPTIMIZE($\mathcal{H}_\theta$)
33:     /* *************************** *End training* *************************** */
34:
35:     /* *generate random samples for C components using fitted $GMM_t$*                    */
36:     $\mathcal{Z}_t^s = $ GENERATE-RANDOM-SAMPLES($GMM_t$) by utilizing Eq.(2) and save it for stage $t+1$

---

# B    TRANSDUCTIVE AND INDUCTIVE EVALUATION

In our main paper, we evaluate our method on the unlabelled data, which are from the train splits of the original datasets. Indeed, the model has seen the data during training, though no labels are used. Here, we further evaluate our method on the test splits of the original datasets, which were not seen by the model during training. In other words, we consider two evaluation protocols, namely, *transductive evaluation* and *inductive evaluation*. In *transductive evaluation*, the model is evaluated on the unlabelled data that has been seen by the model during training, while in *inductive evaluation*, the model is evaluated on the unlabelled data that has not been seen by the model during training.

Since we have reported the transductive evaluation results in the main paper, here, we further include the inductive evaluation results in Table 5, based on the CCD evaluation metric introduced in the main paper. Overall, we can see that our method is more robust to unseen data compared to other models by a large margin as it consistently performs better in the ('All' and 'New') accuracy.

Table 5: Comparison using the CCD evaluation metric under the *inductive* protocol.

|  | Model | Stage 1 ACC (%) | | | Stage 2 ACC (%) | | | Stage 3 ACC (%) | | | Average ACC (%) | | |
|---|---|---|---|---|---|---|---|---|---|---|---|---|---|
|  |  | All | Old | New | All | Old | New | All | Old | New | All | Old | New |
| CIFAR 100 | GCD (Vaze et al., 2022) | 69.68 | 77.96 | 64.29 | 74.43 | 72.38 | 74.82 | 67.80 | 66.67 | 68.00 | 70.64 | 72.34 | 69.04 |
|  | SimGCD (Wen et al., 2023) | 64.03 | 77.96 | 54.29 | 57.86 | 70.95 | 55.36 | 45.25 | 68.57 | 41.17 | 55.71 | 72.97 | 50.27 |
|  | GCD with replay | 68.32 | 74.90 | 63.71 | 54.27 | 63.33 | 52.55 | 59.93 | 70.00 | 58.17 | 60.84 | 69.41 | 58.14 |
|  | SimGCD with replay | 54.29 | 77.55 | 38.00 | 45.80 | 70.00 | 41.18 | 44.54 | 65.24 | 40.92 | 48.21 | 70.93 | 40.03 |
|  | Grow and Merge (Zhang et al., 2022) | 56.05 | 66.53 | 48.71 | 58.63 | 65.71 | 57.27 | 53.62 | 63.33 | 51.92 | 56.10 | 65.19 | 52.63 |
|  | **PromptCCD w/L2P (baseline)** | 77.39 | 73.67 | 80.00 | 78.24 | 68.10 | 80.18 | 70.92 | 58.10 | 73.17 | 75.52 | 66.62 | 77.78 |
|  | **PromptCCD w/GMP (Ours)** | 82.61 | 77.96 | 85.86 | 75.80 | 72.86 | 76.36 | 73.68 | 68.57 | 74.48 | **77.36** | **73.13** | **78.90** |
| ImageNet 100 | GCD (Vaze et al., 2022) | 71.61 | 76.67 | 68.57 | 78.84 | 79.29 | 78.73 | 62.43 | 73.57 | 59.83 | 70.96 | **76.51** | 69.04 |
|  | SimGCD (Wen et al., 2023) | 75.83 | 75.24 | 80.00 | 67.86 | 70.71 | 57.89 | 73.26 | 73.57 | 72.53 | 72.32 | 73.17 | 70.14 |
|  | GCD with replay | 77.14 | 78.57 | 76.29 | 77.68 | 73.57 | 78.73 | 68.38 | 77.14 | 66.33 | 74.40 | 76.43 | 74.78 |
|  | SimGCD with replay | 56.43 | 74.29 | 45.71 | 57.25 | 65.71 | 55.09 | 46.22 | 61.43 | 42.67 | 53.30 | 67.14 | 47.82 |
|  | Grow and Merge (Zhang et al., 2022) | 68.57 | 70.95 | 67.14 | 76.38 | 72.14 | 77.45 | 61.08 | 72.14 | 58.50 | 68.67 | 71.74 | 67.69 |
|  | **PromptCCD w/L2P (baseline)** | 80.00 | 78.10 | 81.14 | 71.30 | 70.71 | 71.45 | 62.84 | 71.43 | 60.83 | 71.38 | 73.41 | 71.14 |
|  | **PromptCCD w/GMP (Ours)** | 77.68 | 76.19 | 78.57 | 75.94 | 73.57 | 76.55 | 71.76 | 75.71 | 70.83 | **75.12** | 75.16 | **75.32** |
| TinyImageNet | GCD (Vaze et al., 2022) | 66.07 | 71.90 | 62.57 | 47.25 | 55.00 | 45.27 | 49.05 | 53.57 | 48.00 | 54.12 | 60.16 | 51.95 |
|  | SimGCD (Wen et al., 2023) | 49.46 | 70.00 | 37.14 | 39.49 | 61.07 | 34.00 | 37.84 | 55.71 | 33.67 | 42.26 | **62.26** | 34.94 |
|  | GCD with replay | 62.55 | 67.50 | 60.71 | 54.64 | 61.07 | 53.00 | 55.31 | 57.86 | 54.42 | 57.50 | 62.04 | 56.04 |
|  | SimGCD with replay | 44.64 | 57.38 | 37.00 | 36.01 | 34.64 | 36.36 | 34.26 | 36.43 | 33.75 | 38.30 | 42.82 | 35.70 |
|  | Grow and Merge (Zhang et al., 2022) | 56.43 | 64.29 | 51.71 | 45.22 | 50.71 | 43.82 | 45.20 | 53.93 | 43.17 | 48.95 | 56.31 | 46.23 |
|  | **PromptCCD w/L2P (baseline)** | 62.14 | 63.33 | 61.43 | 57.46 | 57.14 | 57.55 | 53.45 | 57.14 | 52.58 | 57.68 | 59.20 | 57.19 |
|  | **PromptCCD w/GMP (Ours)** | 62.77 | 71.43 | 57.57 | 57.88 | 58.21 | 57.55 | 57.03 | 55.36 | 57.42 | **59.23** | 61.67 | **57.51** |
| CUB 200 | GCD (Vaze et al., 2022) | 61.09 | 74.63 | 52.21 | 61.86 | 66.17 | 60.21 | 59.54 | 64.40 | 58.04 | 60.83 | 68.23 | 56.82 |
|  | SimGCD (Wen et al., 2023) | 44.06 | 65.00 | 30.07 | 32.37 | 62.86 | 25.91 | 34.38 | 66.43 | 28.13 | 36.94 | 64.76 | 28.03 |
|  | GCD with replay | 60.07 | 80.47 | 53.68 | 46.46 | 77.14 | 39.85 | 60.61 | 72.56 | 55.93 | 55.71 | **76.72** | 49.82 |
|  | SimGCD with replay | 38.82 | 62.86 | 30.79 | 34.88 | 52.14 | 31.21 | 38.68 | 47.86 | 35.10 | 37.46 | 54.29 | 32.37 |
|  | Grow and Merge (Zhang et al., 2022) | 41.60 | 74.22 | 31.37 | 31.77 | 64.29 | 24.77 | 43.96 | 59.93 | 37.71 | 39.11 | 66.15 | 31.28 |
|  | **PromptCCD w/L2P (baseline)** | 52.61 | 76.56 | 45.10 | 53.16 | 72.14 | 49.08 | 57.56 | 66.43 | 54.10 | 54.44 | 71.71 | 49.43 |
|  | **PromptCCD w/GMP (Ours)** | 61.12 | 75.81 | 55.37 | 56.08 | 75.71 | 51.85 | 67.21 | 72.56 | 65.11 | **61.47** | 74.69 | **57.44** |

## C  ADAPTING THE STANDARD GCD METRIC IN EACH TIME STEP OF CCD

In the main paper, when evaluating our method, at each time step $t$, we consider the previously discovered categories as "known" (associated with the pseudo labels obtained by our method), which are included in $D^L$ in the CCD evaluation algorithm. Here, we further consider the case of *not* considering the discovered categories, but only using the actual labelled data $D^L$ as the labelled data at each time step during evaluation following Vaze et al. (2022), as summarized in Algorithm 3 and report the results in Table 6.

Overall, compared with the other models, our method outperforms the other methods in all datasets across all instances ('All' and 'New') accuracy.

---

**Algorithm 3** standard incremental GCD evaluation metric

---

  **Input:** $f(.)$ models for each stage in $\{1, \cdots, T\}$ and datasets $\{D^L, D^U\}$.
  **Output:** The ACC outputs for every stage.
1: Initialize labelled set $D^l$.
2: **for** $t \in \{1, \cdots, T\}$ **do**
3: $\quad\lfloor\; ACC_t$ = SS-KMeans(Model: $f_t(.)$, Labelled set: $D^l$, Unlabelled set: $\{D^u_t\}$)
4: **return** $\{ACC_t \mid t = 1, \ldots, T\}$

---

Table 6: Comparison using the CCD evaluation metric using the adapted GCD evaluation metric in Algorithm 3 under the *transductive* protocol.

| | Model | Stage 1 ACC (%) | | | Stage 2 ACC (%) | | | Stage 3 ACC (%) | | | Average ACC (%) | | |
|---|---|---|---|---|---|---|---|---|---|---|---|---|---|
| | | All | Old | New | All | Old | New | All | Old | New | All | Old | New |
| **CIFAR 100** | GCD (Vaze et al., 2022) | 85.11 | 88.61 | 82.66 | 63.05 | 69.43 | 61.84 | 51.32 | 62.57 | 49.35 | 66.49 | 73.54 | 64.61 |
| | SimGCD (Wen et al., 2023) | 85.70 | 89.68 | 48.29 | 59.85 | 88.86 | 54.31 | 37.86 | 83.43 | 29.88 | 61.14 | **87.32** | 44.16 |
| | GCD with replay | 71.28 | 82.00 | 63.77 | 60.81 | 71.05 | 58.85 | 49.74 | 64.86 | 47.10 | 60.61 | 72.63 | 56.57 |
| | SimGCD with replay | 50.97 | 75.31 | 33.94 | 42.18 | 62.48 | 38.31 | 40.35 | 57.33 | 37.38 | 44.50 | 65.04 | 36.54 |
| | Grow and Merge (Zhang et al., 2022) | 64.77 | 70.49 | 60.77 | 61.25 | 64.00 | 60.73 | 42.91 | 59.24 | 40.05 | 56.31 | 64.58 | 53.85 |
| | **PromptCCD w/L2P (baseline)** | 86.13 | 79.13 | 90.91 | 73.83 | 66.10 | 75.31 | 57.38 | 62.86 | 56.02 | 72.44 | 69.36 | 74.08 |
| | **PromptCCD w/GMP (Ours)** | 87.03 | 88.24 | 84.57 | 83.22 | 76.86 | 78.33 | 58.16 | 69.14 | 56.42 | **76.14** | 78.08 | **74.11** |
| **ImageNet 100** | GCD (Vaze et al., 2022) | 82.45 | 83.51 | 81.71 | 72.49 | 80.95 | 70.87 | 60.04 | 77.52 | 56.98 | 71.66 | 80.66 | 69.85 |
| | SimGCD (Wen et al., 2023) | 83.70 | 84.04 | 81.29 | 69.67 | 77.71 | 41.51 | 70.94 | 76.57 | 57.79 | 74.77 | 79.44 | 60.79 |
| | GCD with replay | 79.75 | 80.82 | 79.00 | 62.23 | 78.86 | 59.05 | 56.78 | 78.48 | 52.98 | 66.25 | 79.39 | 63.68 |
| | SimGCD with replay | 59.78 | 80.00 | 45.63 | 49.60 | 64.57 | 46.75 | 41.22 | 59.05 | 38.10 | 50.20 | 67.87 | 43.49 |
| | Grow and Merge (Zhang et al., 2022) | 75.45 | 76.86 | 74.46 | 65.54 | 75.81 | 63.58 | 56.17 | 76.00 | 52.70 | 65.72 | 76.22 | 63.58 |
| | **PromptCCD w/L2P (baseline)** | 81.95 | 80.69 | 82.83 | 59.25 | 74.38 | 56.36 | 60.62 | 73.71 | 58.33 | 67.27 | 76.26 | 65.84 |
| | **PromptCCD w/GMP (Ours)** | 84.62 | 84.29 | 84.86 | 82.34 | 81.81 | 82.44 | 60.57 | 78.29 | 57.47 | **75.84** | **81.46** | **74.92** |
| **TinyImageNet** | GCD (Vaze et al., 2022) | 65.81 | 70.73 | 62.36 | 58.61 | 61.05 | 56.00 | 48.99 | 54.24 | 48.08 | 57.80 | 62.01 | 55.48 |
| | SimGCD (Wen et al., 2023) | 49.41 | 68.92 | 35.76 | 37.34 | 57.52 | 33.48 | 32.74 | 51.29 | 29.49 | 39.83 | 59.24 | 32.91 |
| | GCD with replay | 63.83 | 65.98 | 62.33 | 55.75 | 61.62 | 54.63 | 49.51 | 58.90 | 47.87 | 56.36 | 62.17 | 54.94 |
| | SimGCD with replay | 41.82 | 52.45 | 34.37 | 34.15 | 32.38 | 34.49 | 31.84 | 26.62 | 32.75 | 35.94 | 37.15 | 33.87 |
| | Grow and Merge (Zhang et al., 2022) | 57.91 | 63.24 | 54.31 | 46.56 | 57.00 | 44.56 | 44.21 | 53.67 | 42.56 | 47.21 | 57.97 | 47.14 |
| | **PromptCCD w/L2P (baseline)** | 69.92 | 64.14 | 73.96 | 63.87 | 59.43 | 64.72 | 50.41 | 55.62 | 49.50 | 61.40 | 59.73 | 62.73 |
| | **PromptCCD w/GMP (Ours)** | 72.75 | 72.65 | 72.81 | 60.94 | 63.14 | 60.52 | 57.95 | 57.24 | 58.07 | **63.88** | **64.34** | **63.80** |
| **CUB200** | GCD (Vaze et al., 2022) | 55.66 | 78.21 | 47.26 | 49.75 | 75.71 | 44.24 | 54.90 | 72.86 | 51.39 | 53.43 | 75.59 | 32.39 |
| | SimGCD (Wen et al., 2023) | 44.06 | 65.00 | 30.07 | 32.37 | 62.86 | 25.91 | 34.38 | 66.43 | 28.13 | 36.94 | 64.76 | 28.04 |
| | GCD with replay | 56.71 | 82.14 | 48.21 | 50.38 | 75.71 | 45.00 | 57.82 | 68.57 | 53.62 | 52.98 | 75.47 | 48.94 |
| | SimGCD with replay | 38.82 | 62.86 | 30.79 | 34.88 | 52.14 | 31.21 | 38.68 | 47.86 | 35.10 | 37.46 | 54.29 | 32.37 |
| | Grow and Merge (Zhang et al., 2022) | 38.64 | 70.71 | 27.92 | 30.50 | 68.57 | 22.42 | 41.88 | 59.64 | 34.96 | 37.01 | 66.31 | 28.43 |
| | **PromptCCD w/L2P (baseline)** | 50.63 | 73.57 | 42.96 | 54.87 | 75.00 | 50.61 | 59.22 | 68.21 | 55.71 | 54.91 | 72.26 | 49.76 |
| | **PromptCCD w/GMP (Ours)** | 59.39 | 82.86 | 51.55 | 56.50 | 79.29 | 51.67 | 64.03 | 77.86 | 58.64 | **59.97** | **80.00** | **53.95** |

# D    DYNAMICALLY ESTIMATING UNKNOWN CLASS NUMBERS DURING LEARNING

When the class numbers in the unlabelled data are unknown, one way to approach this problem is to estimate it in an offline fashion using the method introduced in Vaze et al. (2022) at each time step, as reported in the main paper. In CCD, due to the continual learning nature of the problem, it would be more plausible to estimate the class numbers on the fly automatically. To do so, we draw inspiration from GPC (Zhao et al., 2023), which introduces a semi-supervised GMM (SS-GMM) module that can estimate class numbers dynamically during learning. SS-GMM performs the splitting and merging of clusters during learning by assessing the cluster's compactness and separability using a Markov chain Monte Carlo (MCMC) algorithm. We integrate their class estimation approach seamlessly into our framework, allowing our method to dynamically estimate the unknown class number in each time step when the number is not given.

Specifically, to enable the split-and-merge operation of clusters, each of the Gaussian components of the GMM is further decomposed into two sub-components with $\mu_{c,1}, \mu_{c,2}$ and $\Sigma_{c,1}, \Sigma_{c,2}$ With these parameters, we can calculate the Hastings ratio which measures the compactness and separability of the clusters. The Hastings ratio for splitting a cluster is defined as:

$$H_s = \frac{\Gamma(N_{c,1})h(\mathcal{Z}_{c,1})\Gamma(N_{c,2})h(\mathcal{Z}_{c,2})}{\Gamma(N_c)h(\mathcal{Z}_c)}, \tag{6}$$

where $\Gamma$ is the factorial function, $h$ is the marginal likelihood function of the observed data $\mathcal{Z}$, $\mathcal{Z}_{c,1}$ denotes the data points assigned to the subcluster $\{c, 1\}$, and $N_{c,1}$ is the number of data points in the subcluster $\{c, 1\}$. Note that $H_s$ is in the range of $(0, +\infty)$, thus we will use $p_s = min(1, H_s)$ as a valid probability for performing the splitting operation.

We denote this extension as PromptCCD$^+$, and report the results in Table 7, where we compare four representative models, *i.e.*, (Vaze et al., 2022; Zhang et al., 2022), with our method. We can see that by incorporating the class number estimation module from Zhao et al. (2023) into our model, our method still consistently outperforms other methods by a large margin across the board, demonstrating the superior performance of our approach in the more realistic case when the class numbers are unknown in each time step.

Table 7: Comparison when class numbers are unknown under the *transductive* protocol. Here, we integrate the class number estimation module of GPC to our method to dynamically estimate the class number $C$. The estimated $C$s are applied to all other methods for comparison.

| | Model | Stage 1 ACC (%) | | | Stage 2 ACC (%) | | | Stage 3 ACC (%) | | | Average ACC (%) | | |
|---|---|---|---|---|---|---|---|---|---|---|---|---|---|
| | | All | Old | New | All | Old | New | All | Old | New | All | Old | New |
| **C 100** | Estimated $C$ | $C^{EST}$: 77, $C^{GT}$: 80 | | | $C^{EST}$: 78, $C^{GT}$: 90 | | | $C^{EST}$: 81, $C^{GT}$: 100 | | | | | |
| | GCD (Vaze et al., 2022) | 88.44 | 84.61 | 91.11 | 81.33 | 63.71 | 84.69 | 73.74 | 55.99 | 76.87 | 81.17 | 68.10 | 84.22 |
| | Grow and Merge (Zhang et al., 2022) | 58.17 | 68.82 | 50.71 | 56.93 | 58.67 | 56.60 | 54.92 | 54.38 | 55.02 | 56.67 | 60.62 | 54.11 |
| | **PromptCCD$^+$ w/GMP (Ours)** | 90.82 | 84.82 | 95.03 | 81.85 | 68.67 | 84.36 | 76.18 | 60.76 | 78.88 | **82.95** | **71.42** | **86.09** |
| **IN 100** | Estimated $C$ | $C^{EST}$: 73, $C^{GT}$: 80 | | | $C^{EST}$: 73, $C^{GT}$: 90 | | | $C^{EST}$: 83, $C^{GT}$: 100 | | | | | |
| | GCD (Vaze et al., 2022) | 78.69 | 79.96 | 77.80 | 69.07 | 68.10 | 69.52 | 80.71 | 66.86 | 83.13 | 76.16 | 71.64 | 76.82 |
| | Grow and Merge (Zhang et al., 2022) | 53.70 | 75.84 | 38.20 | 41.13 | 71.81 | 35.27 | 64.70 | 69.24 | 63.90 | 53.18 | 72.29 | 45.79 |
| | **PromptCCD$^+$ w/GMP (Ours)** | 82.82 | 80.61 | 83.71 | 67.98 | 71.62 | 67.42 | 86.27 | 70.95 | 88.95 | **79.02** | **74.39** | **80.03** |
| **Tiny 200** | Estimated $C$ | $C^{EST}$: 158, $C^{GT}$: 160 | | | $C^{EST}$: 164, $C^{GT}$: 180 | | | $C^{EST}$: 168, $C^{GT}$: 200 | | | | | |
| | GCD (Vaze et al., 2022) | 69.97 | 70.33 | 69.73 | 56.71 | 56.71 | 56.71 | 52.19 | 51.86 | 52.25 | 59.62 | 59.63 | 59.56 |
| | Grow and Merge (Zhang et al., 2022) | 55.82 | 62.20 | 51.34 | 46.88 | 53.62 | 45.59 | 50.38 | 49.81 | 50.48 | 51.03 | 55.21 | 49.14 |
| | **PromptCCD$^+$ w/GMP (Ours)** | 73.72 | 72.22 | 74.77 | 59.55 | 59.24 | 59.61 | 63.57 | 52.95 | 65.43 | **65.61** | **61.47** | **66.60** |
| **CUB 200** | Estimated $C$ | $C^{EST}$: 163, $C^{GT}$: 160 | | | $C^{EST}$: 172, $C^{GT}$: 180 | | | $C^{EST}$: 175, $C^{GT}$: 200 | | | | | |
| | GCD (Vaze et al., 2022) | 59.23 | 78.57 | 46.30 | 47.00 | 70.71 | 41.97 | 49.65 | 62.86 | 47.08 | 51.96 | 70.71 | 45.12 |
| | Grow and Merge (Zhang et al., 2022) | 42.49 | 63.57 | 28.40 | 27.50 | 61.43 | 20.30 | 40.91 | 64.29 | 36.35 | 36.97 | 63.10 | 28.35 |
| | **PromptCCD$^+$ w/GMP (Ours)** | 53.65 | 74.29 | 39.62 | 44.00 | 77.86 | 38.62 | 62.94 | 67.14 | 62.12 | **53.53** | **73.10** | **46.79** |

# E  IMPLEMENTATION DETAILS FOR AUGMENTING GROW AND MERGE WITH VIT

As the most relevant work Grow and Merge (Zhang et al., 2022) uses ResNet18 (He et al., 2016) as the backbone and the Momentum Contrast (MoCo) (He et al., 2020) for representation learning, to have a fair comparison, we augment Grow and Merge from two aspects, the pretraining strategy and the dual branch network (static and dynamic branch), leveraging the more powerful ViT backbone. First, we change the pretraining strategy MoCo to the joint supervised and unsupervised contrastive learning with DINO features. Second, for the dual branch network in Zhang et al. (2022), originally, the ResNet18 is divided into several layers (excluding the fully connected layers) where before the last layer, Grow and Merge divides the last layer into two branches, *i.e.*, the static branch and the dynamic branch. By design, The static branch is the backbone's last layer, while the dynamic branch consists of several branches of $T-1$ layers, where $T$ is the number of stages. To maintain this design, we accordingly implement a dual-branch architecture network based on ViT backbone. Given that ViT backbone consists of several blocks, we freeze all blocks except the last block as the static branch. Moreover, before the last block, we add another $T-1$ blocks as the dynamic branches used exclusively for each stage $t$. All the rest designs are the same as Zhang et al. (2022). At $t = 0$, *i.e.*, during the initial stage, we optimize the static branch, and at $t > 0$, we freeze the static branch and perform static-dynamic distillation while optimizing the dynamic branch $t$ for novel class discovery following Grow and Merge.

In Table 8, we compare our method with the improved Grow and Merge under both *transductive* and *inductive* evaluation protocols, using the CCD evaluation metric. Our *improved* Grow and Merge significantly outperforms the original implementation, leading to a fair comparison with our method. However, our method still notably outperforms the improved Grow and Merge, demonstrating the superiority of our approach. Moreover, to investigate the performance of Grow and Merge in the initial learning stage, we further experiment with GM CCD training with only SSL methods in the initial stage shown in the third row of *Transductive Evaluation*. This change leads to a small improvement of around $1\%$ for GM, with an overall ACC of $58.37\%$. However, our method obtains an overall ACC of $84.22\%$, which is still substantially better.

**Note**: For experiments in main paper, we compare our model with the *improved* Grow and Merge.

Table 8: Comparison different Grow and Merge implementations with our method on CIFAR 100.

| Model | Stage 1 ACC (%) | | | Stage 2 ACC (%) | | | Stage 3 ACC (%) | | | Average ACC (%) | | |
|---|---|---|---|---|---|---|---|---|---|---|---|---|
| | All | Old | New | All | Old | New | All | Old | New | All | Old | New |
| *Transductive Evaluation* | | | | | | | | | | | | |
| GM (Zhang et al., 2022) | 22.91 | 30.20 | 17.80 | 21.47 | 25.71 | 20.65 | 24.91 | 24.00 | 27.25 | 23.10 | 26.64 | 21.90 |
| GM (*improved*) | 64.77 | 70.49 | 60.77 | 58.31 | 62.95 | 57.42 | 48.82 | 56.00 | 47.57 | **57.30** | **63.14** | **55.25** |
| GM w/GCD initial pretrained | 64.74 | 70.49 | 60.71 | 61.25 | 64.00 | 60.73 | 49.13 | 57.52 | 47.67 | 58.37 | 64.00 | 56.37 |
| PromptCCD w/GMP | 90.20 | 90.73 | 92.51 | 85.83 | 75.62 | 87.78 | 76.64 | 67.14 | 78.30 | **84.22** | **77.83** | **86.20** |
| *Inductive Evaluation* | | | | | | | | | | | | |
| GM (Zhang et al., 2022) | 38.32 | 60.61 | 22.71 | 29.62 | 60.48 | 23.73 | 31.91 | 60.95 | 26.83 | 33.28 | 60.68 | 24.42 |
| GM (*improved*) | 56.05 | 66.53 | 48.71 | 58.63 | 65.71 | 57.27 | 53.62 | 63.33 | 51.92 | **56.10** | **65.19** | **52.63** |
| PromptCCD w/GMP | 82.61 | 77.96 | 85.86 | 75.80 | 72.86 | 76.36 | 73.68 | 68.57 | 74.48 | **77.36** | **73.13** | **78.90** |

# F    COMPARISON WITH RECENT WORKS ON CCD TASK

In this section, we further experiment on CCD concurrent works, which are *"Proxy Anchor-based Unsupervised Learning for Continuous Generalized Category Discovery"* (Kim et al., 2023) and *"MetaGCD: Learning to Continually Learn in Generalized Category Discovery"* (Wu et al., 2023). We further compare our method with these methods in Table. 9 and Table. 10 following their settings *i.e.*, pretrained model, data distribution, evaluation protocols, dataset. Overall, our model still outperforms their model in all metrics.

Table 9: Comparison with Proxy-Anchor based unsupervised learning Incremental GCD (PA) (Kim et al., 2023) Table 4, DINO ViT-B-16 experiments on CUB200. For experiment settings and evaluation metrics, please refer to the original paper section 4.2.

| Model | $\mathcal{M}_{all}\uparrow$ | $\mathcal{M}_o\uparrow$ | $\mathcal{M}_f\downarrow$ | $\mathcal{M}_d\uparrow$ |
|---|---|---|---|---|
| GCD (Vaze et al., 2022) | 62.70 | 71.40 | 09.57 | 56.01 |
| Grow and Merge (Zhang et al., 2022) | 42.12 | 60.21 | 23.24 | 27.63 |
| PA (Kim et al., 2023) | 72.51 | 74.28 | 09.49 | 65.60 |
| PromptCCD w/GMP (Ours) | **76.23** | **78.44** | **06.07** | **74.46** |

Table 10: Comparison with MetaGCD (Wu et al., 2023) on CIFAR100. For experimental settings and evaluation metrics, please refer to the original paper section 4.

| Model | Stage 1 ACC (%) | | | Stage 2 ACC (%) | | | Stage 3 ACC (%) | | | Stage 4 ACC (%) | | | Average ACC (%) | | |
|---|---|---|---|---|---|---|---|---|---|---|---|---|---|---|---|
| | All | Old | New | All | Old | New | All | Old | New | All | Old | New | All | Old | New |
| MetaGCD (Wu et al., 2023) | 78.96 | 79.36 | 72.60 | 78.67 | 79.41 | 66.81 | 76.06 | 78.20 | 64.87 | 74.56 | 77.60 | 61.14 | 77.06 | 78.64 | 66.35 |
| **PromptCCD w/GMP (Ours)** | 90.06 | 90.50 | 89.47 | 82.67 | 88.80 | 76.23 | 81.48 | 84.60 | 78.80 | 70.30 | 75.87 | 67.64 | **81.13** | **84.94** | **78.04** |

# G ANALYSIS ON DIFFERENT CLASS SPLIT RATIO

In the main paper, we follow the same data split ratio introduced by Grow and Merge (Zhang et al., 2022) *i.e.*, 7:1:1:1, which contains 4 learning stages where the first stage is the initial learning on labelled data. To further mimic the real-world scenario, which is characterized by an abrupt increase or decrease in the number of classes of each stage, we experiment on another 3 different data split scenarios *i.e.*, scenario 1 (4:2:2:2) where the number of the unseen classes is greater than the seen class, scenario 2 (4:3:2:1) where the number of the unseen classes is decreasing for each stage, and finally scenario 3 (1:2:3:4) where the number of the unseen class is increasing for each stage. As shown in Table. 11, we compare our model with 2 other representative models *i.e.*, GCD and Grow and Merge on the CIFAR100 dataset. In all these challenging cases, our model still substantially outperforms other methods across the board.

Table 11: Experiments on different class split ratio on CIFAR 100.

| Model | Stage 1 ACC (%) | | | Stage 2 ACC (%) | | | Stage 3 ACC (%) | | | Average ACC (%) | | |
|---|---|---|---|---|---|---|---|---|---|---|---|---|
| | All | Old | New | All | Old | New | All | Old | New | All | Old | New |
| *Class Split Ratio: 4:2:2:2* | | | | | | | | | | | | |
| GCD (Vaze et al., 2022) | 78.25 | 55.36 | 80.54 | 65.79 | 39.83 | 66.98 | 38.72 | 39.83 | 38.50 | 60.92 | 45.01 | 62.01 |
| Grow and Merge (Zhang et al., 2022) | 51.17 | 41.86 | 52.10 | 45.90 | 31.50 | 46.57 | 34.72 | 45.17 | 32.63 | 43.93 | 39.51 | 43.77 |
| PromptCCD w/GMP | 78.53 | 50.50 | 81.34 | 74.22 | 59.67 | 74.89 | 52.64 | 43.50 | 54.47 | **68.46** | **51.22** | **70.23** |
| *Class Split Ratio: 4:3:2:1 (decreasing)* | | | | | | | | | | | | |
| GCD (Vaze et al., 2022) | 58.62 | 59.79 | 58.50 | 51.99 | 42.17 | 52.44 | 40.69 | 40.83 | 40.67 | 50.43 | 47.59 | 50.54 |
| Grow and Merge (Zhang et al., 2022) | 41.89 | 52.83 | 39.70 | 44.25 | 42.83 | 44.32 | 34.97 | 35.50 | 34.87 | 40.37 | 43.72 | 39.63 |
| PromptCCD w/GMP | 57.10 | 61.14 | 55.70 | 64.10 | 51.00 | 64.70 | 47.67 | 38.67 | 49.47 | **56.29** | **50.27** | **56.62** |
| *Class Split Ratio: 1:2:3:4 (increasing)* | | | | | | | | | | | | |
| GCD (Vaze et al., 2022) | 52.89 | 63.21 | 51.86 | 53.94 | 53.67 | 53.95 | 45.49 | 33.00 | 45.21 | 50.77 | 49.96 | 50.34 |
| Grow and Merge (Zhang et al., 2022) | 50.40 | 44.64 | 50.98 | 44.48 | 36.33 | 44.85 | 41.89 | 52.83 | 39.70 | 45.59 | 44.60 | 45.18 |
| PromptCCD w/GMP | 50.21 | 63.57 | 48.88 | 49.96 | 60.50 | 49.47 | 57.00 | 58.00 | 56.80 | **52.39** | **60.69** | **51.72** |

# H   WHY FINETUNE THE FINAL BLOCK OF DINO FOR CCD?

We analyze the number of learning parameters for each compared model and explain why the final block of our backbone is fine-tuned. Our motivation is to repurpose SSL vision foundation models for CCD. We choose DINO (Caron et al., 2021) as our vision foundation model to tackle CCD. DINO is a transformer-based vision foundation model pretrained on ImageNet 1K (Russakovsky et al., 2015) with a resolution of $224 * 224$ pixels. The model is trained in a self-supervised manner (no label information) with around $86M$ parameters. SSL models have been widely adopted and justified in both NCD (Han et al., 2021) and almost all GCD literature so far. Thus, we use DINO's self-supervised pretrained model for all compared models. We finetune the final block of its backbone and report the number of learnable parameters for each model in Table. 12. Our model's learnable parameters consist of two parts: the final block of the backbone and the parameter from GMP's GMM. The latter only accommodates $\{(2 * |z| + 1) * C\}$ parameters, where $C$ is the number of components, $|z|$ is the feature size of the $z$[CLS] tokens, which in this case is 768. Compared with PromptCCD w/{ L2P, DP }, our model's learnable parameters are only $0.33\%$ higher when $C = 100$, which is still efficient.

Table 12: Information on learnable parameters for each compared model

| Model | Learnable Parameters | $\approx$ Total Parameters |
|---|---|---|
| Orca (Cao et al., 2022) | $7.1M$ $f_b$; $6.5M$ Classification head | $13.6M$ |
| GCD (Vaze et al., 2022) | $7.1M$ $f_b$; $23.1M$ $\phi$ | $30.2M$ |
| SimGCD (Wen et al., 2023) | $7.1M$ $f_b$; $6.5M$ Classification head | $13.6M$ |
| GM (Zhang et al., 2022) | $7.1M$ $f_b$; $23.1M$ $\phi$; $0.031M$ Cluster head | $30.2M$ |
| PromptCCD w/L2P | $7.1M$ $f_b$; $23.1M$ $\phi$; $0.046M$ L2P | $30.2M$ |
| PromptCCD w/DP | $7.1M$ $f_b$; $23.1M$ $\phi$; $0.045M$ DP | $30.2M$ |
| PromptCCD w/GMP (Ours) | $7.1M$ $f_b$; $23.1M$ $\phi$; $\{1537 * C\}$ GMP | $30.3M$ @ $C = 100$ |

L2P and DualPrompt (Wang et al., 2022b;a) are prompt-based models designed for supervised continual learning task. Both models freeze the backbone model and train the linear classifier in a supervised manner. Our CCD model $\mathcal{H}_\theta : \{\phi, f_\theta\}$ consists of $\phi$, an MLP projection head, and $f_\theta : \{f_e, f_b\}$ a transformer-based feature backbone that includes an input embedding layer $f_e$ and self-attention blocks $f_b$. During training, we optimize both the final block of $f_b$ and the projection head $\phi$. $\phi$ is used to optimize the model in a self-supervised manner by projecting high-dimensional features to a simpler dimension before calculating the loss. Thus, it is not possible to completely freeze the backbone as there will be no learnable parameters left for inference. To justify the reason to fine-tune the final block of the backbone, we experiment with two frozen DINO models. The first model is the default frozen DINO backbone with no prompt module. For this model, we do not perform any training strategy and directly use it to extract $z$[CLS] tokens. The second model is the frozen DINO backbone coupled with a learnable L2P prompt pool. For this model, we follow the exact training procedure similar to the baseline model but keep the backbone frozen. We compare these two frozen models with both our fine-tuned baseline and proposed models as shown in Table. 13. By comparing the performance of the frozen models and the fine-tuned models, we can see that our fine-tuned model substantially outperforms the frozen models. Furthermore, if we observe the performance of our fine-tuned models on C100 and CUB200 datasets, we can see that our models generalized better to datasets that the DINO foundation model has not encountered before which further justifies the design choice of our method for CCD.

Table 13: Comparison between the fully frozen models and the fine tuned (final block) models.

| | Model | Stage 1 ACC (%) | | | Stage 2 ACC (%) | | | Stage 3 ACC (%) | | | Average ACC (%) | | |
|---|---|---|---|---|---|---|---|---|---|---|---|---|---|
| | | All | Old | New | All | Old | New | All | Old | New | All | Old | New |
| C100 | Frozen DINO (Caron et al., 2021) | 64.87 | 71.43 | 60.29 | 55.42 | 66.67 | 53.27 | 49.08 | 66.19 | 46.08 | 56.45 | 68.10 | 53.21 |
| | Frozen DINO w/L2P | 65.08 | 73.39 | 59.26 | 55.43 | 64.10 | 53.69 | 49.52 | 67.05 | 46.17 | 56.67 | 68.18 | 53.04 |
| | **PromptCCD w/L2P** | 86.77 | 79.76 | 91.69 | 85.05 | 64.10 | 89.05 | 73.45 | 56.95 | 76.33 | 81.75 | 66.94 | 85.69 |
| | **PromptCCD w/GMP (Ours)** | 90.20 | 90.73 | 92.51 | 85.83 | 75.62 | 87.78 | 76.64 | 67.14 | 78.30 | **84.22** | **77.83** | **86.20** |
| IN100 | Frozen DINO (Caron et al., 2021) | 68.75 | 71.90 | 66.86 | 70.43 | 73.57 | 69.64 | 62.57 | 74.29 | 59.83 | 67.25 | 73.25 | 65.44 |
| | Frozen DINO w/L2P | 76.71 | 77.80 | 75.77 | 64.33 | 67.05 | 63.24 | 63.70 | 76.86 | 61.40 | 68.24 | 73.90 | 66.80 |
| | **PromptCCD w/L2P** | 81.95 | 80.69 | 82.83 | 65.77 | 73.81 | 64.24 | 66.52 | 73.05 | 65.38 | 71.41 | 75.85 | 70.82 |
| | **PromptCCD w/GMP (Ours)** | 84.62 | 84.29 | 84.86 | 80.06 | 79.62 | 80.15 | 82.75 | 77.62 | 83.65 | **82.47** | **80.51** | **82.88** |
| Tiny200 | Frozen DINO (Caron et al., 2021) | 55.71 | 65.00 | 52.00 | 45.80 | 56.79 | 43.00 | 46.23 | 55.85 | 42.83 | 49.25 | 59.21 | 45.94 |
| | Frozen DINO w/L2P | 62.02 | 66.31 | 59.01 | 52.20 | 61.00 | 50.52 | 46.42 | 54.81 | 44.95 | 53.54 | 60.70 | 51.49 |
| | **PromptCCD w/L2P** | 69.92 | 64.14 | 73.96 | 68.69 | 59.76 | 70.40 | 56.96 | 52.81 | 57.68 | 65.19 | 58.90 | 67.34 |
| | **PromptCCD w/GMP (Ours)** | 72.75 | 72.65 | 72.81 | 62.01 | 59.71 | 62.45 | 65.16 | 56.76 | 67.19 | **66.64** | **63.04** | **67.48** |
| CUB200 | Frozen DINO (Caron et al., 2021) | 41.60 | 74.22 | 31.37 | 31.27 | 68.57 | 23.23 | 44.77 | 62.09 | 37.99 | 39.21 | 68.29 | 30.86 |
| | Frozen DINO w/L2P | 40.25 | 75.71 | 28.40 | 30.63 | 73.57 | 21.52 | 45.99 | 65.36 | 38.44 | 38.95 | 71.54 | 29.45 |
| | **PromptCCD w/L2P** | 50.63 | 73.57 | 42.96 | 52.38 | 72.14 | 48.18 | 60.12 | 69.29 | 56.55 | 54.38 | 71.67 | 49.23 |
| | **PromptCCD w/GMP (Ours)** | 59.39 | 82.86 | 51.55 | 56.25 | 79.29 | 51.36 | 65.43 | 73.21 | 62.40 | **60.36** | **78.45** | **55.10** |

# I ABLATION STUDIES ON PROMPT MODULE DESIGN

We compare the clustering performance of our fitted GMM acquired after training our proposed PromptCCD w/GMP with standard K-means clustering algorithm using labelled data $D_L$ belonging to the CIFAR100 dataset. We use standard metrics such as normalized mutual information (NMI), adjusted rand index (ARI), purity, and cluster accuracy to evaluate the clustering performance. We also compare other features such as from GCD (Vaze et al., 2022) (trained in CCD setup) and frozen DINO (Caron et al., 2021) backbones, in addition to $z$[CLS] features from our proposed model. The graph in Figure 6 shows that our fitted GMM with PromptCCD features outperforms the other models in all four metrics.

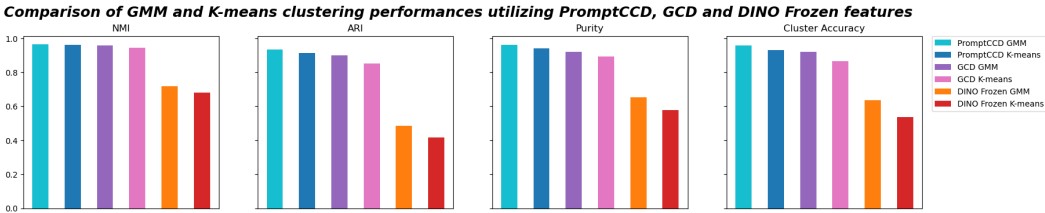

Figure 6: Comparison between GMM and K-means clustering performances.

In addition, we visualize the confusion matrix between the actual true class index from $D_L$ and the predicted cluster assignment by our GMM for both coarse and fine-grained datasets as shown in Figures 7 and 8 respectively. From these figures, we can see that our GMM clustering quality is decent as each sample belonging to the same class is mostly grouped. Thus, from these observations, we can conclude that our GMM learns class prototype which makes our hypothesis true. This means that by prompting GMP prompts, GMP guides the model by giving some top-k class prototype (to accommodate the absence of label information) which guides/enforces the model about the criteria / to look at (information/features that the model should care about) when making the decision to discover a category.

Table 14: Study on the effectiveness of top-k prompts in CCD compared with randomly picked prompts (random-k) from GMM.

| | Model | Stage 1 ACC (%) | | | Stage 2 ACC (%) | | | Stage 3 ACC (%) | | | Average ACC (%) | | |
|---|---|---|---|---|---|---|---|---|---|---|---|---|---|
| | | All | Old | New | All | Old | New | All | Old | New | All | Old | New |
| C100 | PromptCCD w/GMP (random-k) | 85.80 | 87.67 | 84.49 | 70.49 | 72.29 | 70.15 | 57.80 | 67.24 | 56.15 | 71.36 | 75.73 | 70.26 |
| | PromptCCD w/GMP (top-k) (Ours) | 90.20 | 90.73 | 92.51 | 85.83 | 75.62 | 87.78 | 76.64 | 67.14 | 78.30 | **84.22** | **77.83** | **86.20** |
| IN100 | PromptCCD w/GMP (random-k) | 78.47 | 82.61 | 75.57 | 74.41 | 79.43 | 73.45 | 58.82 | 77.52 | 55.55 | 70.46 | 79.85 | 68.19 |
| | PromptCCD w/GMP (top-k) (Ours) | 84.62 | 84.29 | 84.86 | 80.06 | 79.62 | 80.15 | 82.75 | 77.62 | 83.65 | **82.47** | **80.51** | **82.88** |
| Tiny | PromptCCD w/GMP (random-k) | 59.39 | 70.71 | 54.86 | 47.42 | 56.43 | 44.61 | 47.84 | 49.49 | 47.30 | 51.55 | 58.88 | 48.92 |
| | PromptCCD w/GMP (top-k) (Ours) | 72.75 | 72.65 | 72.81 | 62.01 | 59.71 | 62.45 | 65.16 | 56.76 | 67.19 | **66.64** | **63.04** | **67.48** |
| CUB | PromptCCD w/GMP (random-k) | 39.91 | 53.93 | 30.55 | 27.00 | 55.00 | 21.06 | 33.57 | 55.00 | 29.39 | 33.49 | 54.64 | 27.00 |
| | PromptCCD w/GMP (top-k) (Ours) | 59.39 | 82.86 | 51.55 | 56.25 | 79.29 | 51.36 | 65.43 | 73.21 | 62.40 | **60.36** | **78.45** | **55.10** |

Moreover, as shown in Table 14, we experiment on different datasets where instead of taking the "top-k" mean components/prompts, we vary the prompts "random-k" to observe the effect of difference prompt relevancy towards overall CCD performance. We can see that varying the prompt hurts the model's performance, especially for the "NEW" ACC *i.e.*, novel categories.

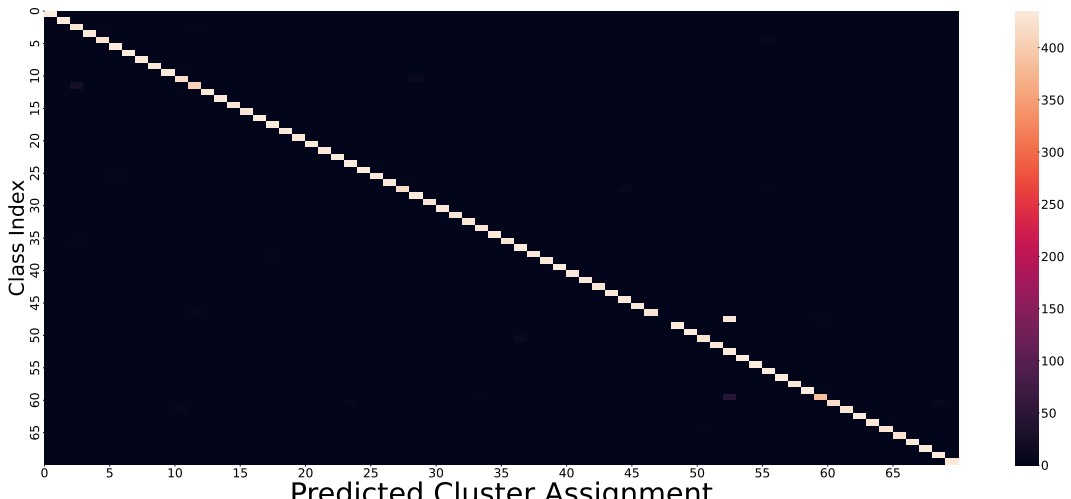

Figure 7: Confusion matrix on PromptCCD GMP's GMM clustering performance on CIFAR100 labelled set $D_L$. Please note that GMM assigned a sample to a cluster and we re-assigned the cluster by relocating large values on the diagonal line. Note that our GMM failed to seperate class id: 47 "Oak Tree" and class id: 52 "Maple Tree".

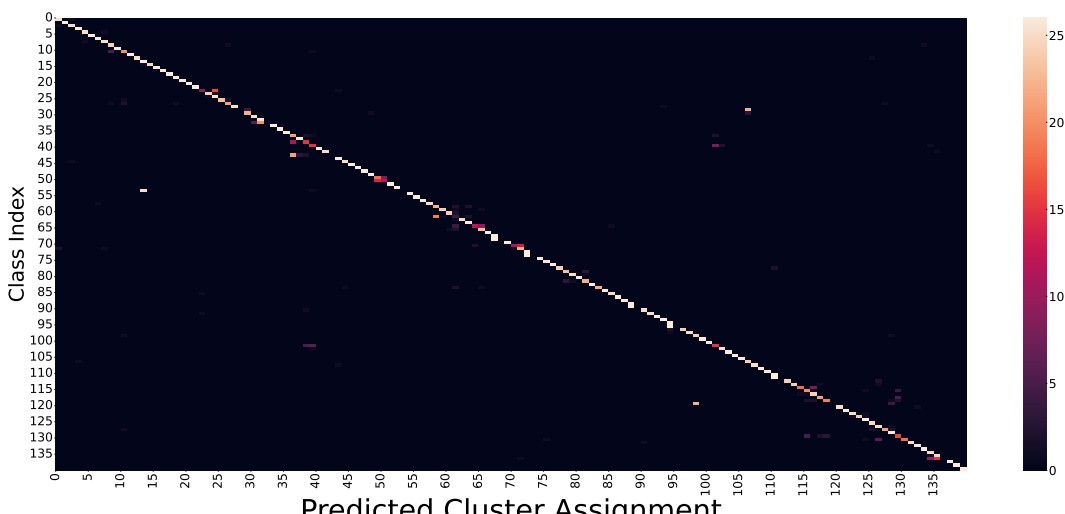

Figure 8: Confusion matrix on PromptCCD GMP's GMM clustering performance on CUB200 labelled set $D_L$. Please note that GMM assigned a sample to a cluster and we re-assigned the cluster by relocating large values on the diagonal line.

# J    ABLATION STUDIES ON THE INITIAL STEPS OF DISCOVERY

Here, we analyze the contribution of each component in the first timestep. The tables can be seen below, which consist of the ablation studies on representation learning, the effectiveness of using the GMP module, and finally, the ablation study on the GMP module itself. As shown in the Table. 15, we show the performance of Grow and Merge (Zhang et al., 2022) and our proposed method. Particularly for Grow and Merge, we study the model's representation learning techniques by ablating its pretrained backbone (ResNet18 vs DINO-ViT, all pretrained ImageNet1K) and the feature training (finetuning) methods (CL vs MoCo). CL and MoCo differ because the latter requires an EMA model.

As shown in Table. 15, by comparing rows 1,2, and 3, we can see that row 3 *i.e.*, Grow and Merge pretrained with DINO model + trained with unsupervised contrastive method (CL), yields the best-performing results with an 'ALL' accuracy performance of 64.77

Moreover, we also show the experiment when GMP is used or not used in a model (see Table. 16). As GMP is simple and can be integrated easily into a model, we experiment with GMP with Grow and Merge, as shown in the second table. We see that the GMP module improves the performance of GM substantially, but our PromptCCD w/GMP module still outperforms GM w/our GMP.

Lastly, to verify the effectiveness of our proposed method, we also show our model ablation study in the first timestep. As shown in Table. 17, 5 prompts and 20 GMM samples lead to the best performance of stage 1. However, the overall performance across stages, as shown in the main paper's ablation studies, shows that 5 prompts and 100 samples lead to the best "ALL" accuracy.

Table 15: Ablation study on the effectiveness of pretraining techniques in the initial step

| Row ID | Model | Backbone | Learning Methods | All | Old | New |
|--------|-------|----------|------------------|-----|-----|-----|
| 1 | GM | ResNet18 | MoCo | 22.91 | 30.20 | 17.80 |
| 2 | GM | ResNet18 | CL | 25.80 | 33.67 | 20.29 |
| 3 | GM | DINO | MoCo | 64.74 | 70.49 | 60.71 |
| 4 | GM | DINO | CL | 64.77 | 70.49 | 60.77 |
| 5 | GCD | DINO | CL | 85.11 | 88.61 | 82.66 |
| 4 | PromptCCD w/GMP | DINO | CL | **90.20** | **90.73** | **92.51** |

Table 16: Ablation study on the effectiveness of GMP module in the initial step

| Row ID | Model | Prompt | All | Old | New |
|--------|-------|--------|-----|-----|-----|
| 1 | GM | w/o GMP | 64.77 | 70.49 | 60.77 |
| 2 | GM | w/ GMP | 77.53 | 82.86 | 73.80 |
| 3 | PromptCCD | w/o GMP | 85.11 | 88.61 | 82.66 |
| 4 | PromptCCD | w/ GMP | **90.20** | **90.73** | **92.51** |

Table 17: Ablation study on different components of our approach in the initial step

| No. Prompt | No. GMM Sampling | Sup.Con | All | Old | New |
|-----------|------------------|---------|-----|-----|-----|
| 0 prompt | 0 sample | ✓ | 85.11 | 88.61 | 82.66 |
| 5 prompts | 100 samples | ✗ | 61.16 | 70.78 | 54.43 |
| 2 prompts | 100 samples | ✓ | 88.34 | 87.55 | 88.89 |
| 10 prompts | 100 samples | ✓ | 79.71 | 82.61 | 77.69 |
| 5 prompts | 0 samples | ✓ | 88.96 | 86.33 | 90.80 |
| 5 prompts | 20 samples | ✓ | 91.58 | 88.78 | 93.54 |
| 5 prompts | 100 samples | ✓ | **90.20** | 90.73 | 92.51 |

# K    ADDITIONAL ABLATION STUDIES

We provide another ablation table on the number of prompts and samples in Table. 18, where we keep all covariance of GMMs spherical. The results show that 5 prompts and 100 samples lead to the best "ALL" accuracy. With the same number of prompts, more samples lead to better results. With the same number of samples, we found that 5 prompts work the best. Moreover, we would like to clarify that our method's supervised contrastive learning module is essential for learning general representations for both labelled and unlabelled categories. Using this loss in the category discovery process is a common practice, such as SimGCD (Wen et al., 2023), GCD (Vaze et al., 2022), and IGCD (Zhao & Mac Aodha, 2023). On the other hand, our proposed GMP module prompts the CCD model to avoid forgetting and guiding the representation learning process. From our ablation studies in the main paper and Table. 18, we can see that without any of the GMP modules or the supervised contrastive learning module, the performance drops, and the combination of both modules achieves the best result, indicating that the two modules are orthogonal to each other.

Table 18: Ablation study on different components of our approach

| No. Prompt | No. GMM Sampling | Sup.Con | C100 Avg ACC (%) | | | CUB200 Avg ACC (%) | | |
|---|---|---|---|---|---|---|---|---|
| | | | All | Old | New | All | Old | New |
| 0 prompt | 0 sample | ✓ | 73.62 | 73.69 | 73.02 | 55.46 | 74.16 | 48.79 |
| 5 prompts | 100 samples | ✗ | 57.27 | 63.34 | 54.51 | 33.13 | 53.57 | 26.62 |
| 2 prompts | 100 samples | ✓ | 82.29 | 75.66 | 83.64 | 59.88 | 72.50 | 54.60 |
| 10 prompts | 100 samples | ✓ | 74.65 | 75.54 | 74.04 | 56.86 | 74.92 | 49.59 |
| 5 prompts | 0 samples | ✓ | 79.67 | 76.42 | 80.72 | 57.58 | 75.71 | 51.06 |
| 5 prompts | 20 samples | ✓ | 80.30 | 76.49 | 81.49 | 59.24 | 77.74 | 53.11 |
| 5 prompts | 100 samples | ✓ | **84.22** | 77.83 | 86.20 | **60.06** | 75.84 | 54.01 |

# L    MORE QUALITATIVE RESULTS

We further visualize the feature representation generated by our method on ImageNet100 (Russakovsky et al., 2015), TinyImageNet (Le & Yang, 2015), and CUB 200 (Wah et al., 2011) datasets , using t-SNE algorithm (Van der Maaten & Hinton, 2008) to project the high-dimensional features of $\{D^l, D_t^u\}$ in each stage into low-dimensional space. The qualitative visualization can be seen in Fig. 9; nodes of the same colour indicate that the instances belong to the same category. Moreover, for stage $t > 0$, we only highlight the feature's node belonging to unknown categories. It is observed that across stages and datasets, our cluster features are discriminative.

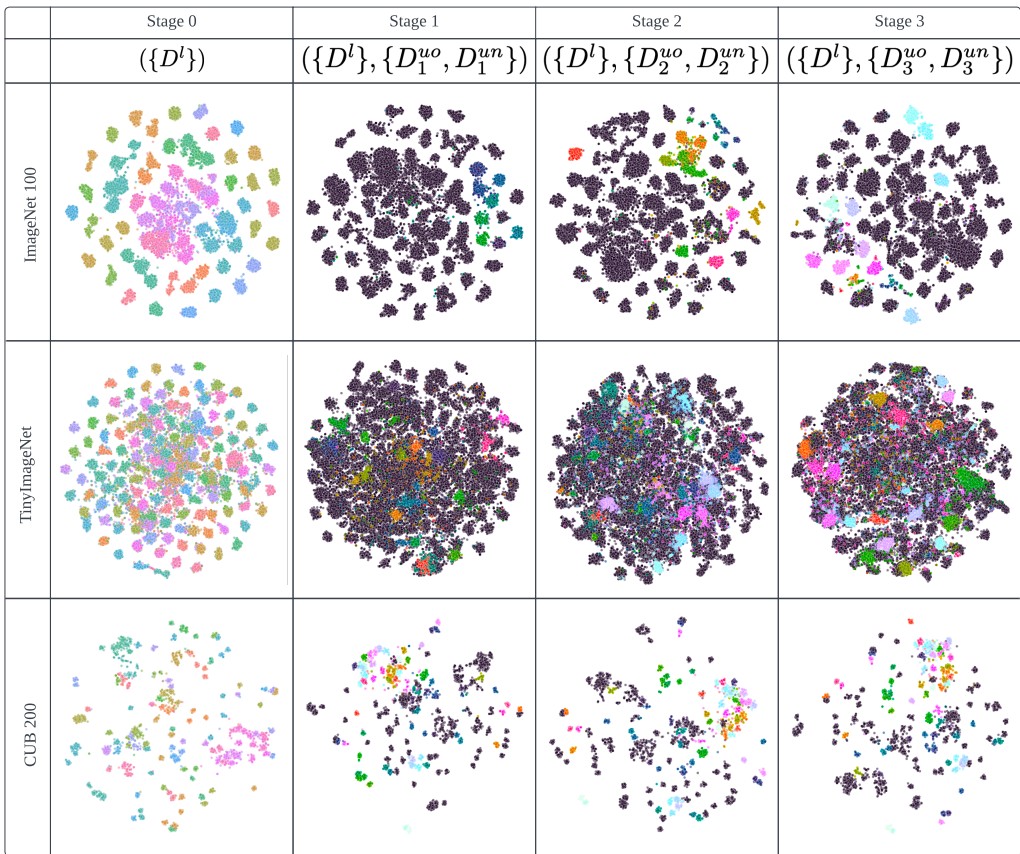

Figure 9: TSNE visualization of features of ImageNet100 , TinyImageNet, and CUB 200 datasets learned by our PromptCCD w/GMP for every stage.

## M    BROADER IMPACTS AND LIMITATIONS

Category discovery technologies significantly impact various industries and applications, such as drug discovery and materials discovery. Our proposed framework has been shown to reduce forgetting while being robust enough to discover new classes. However, there may be some potential negative social impacts, such as when the model learns bad prior knowledge or the data contains unwanted bias, leading to misinformation in society. Currently, we still do not have a mechanism to prevent such situations from happening. Therefore, having proper priors and managing data distribution is important to prevent the model from misclassifying objects. Additionally, like other efforts on handling sequential unlabelled data, our system may accumulate errors over time as we do not have any specific regulation when dealing with longer time steps and potential categories with few samples at a given time step. Thus, more efforts are still needed to improve the capability of AI systems to learn from unlabelled data reliably.

