# OpenReview forum: "PromptCCD: Learning Gaussian Mixture Prompt Pool for Continual Category Discovery"
_ICLR.cc/2024/Conference — Submitted to ICLR 2024_

### Official Review · Reviewer_si3G · 2023-10-31

**Soundness:** 3 good
**Presentation:** 3 good
**Contribution:** 2 fair
**Rating:** 3
**Confidence:** 5

**Summary:**

This paper addresses the challenge of continual category discovery (CCD). CCD consists of two stages: the first stage involves training on a labeled dataset with known categories, while the second stage encompasses a continual setting where unlabeled data arrives sequentially. In each continual session, the objective is to identify new categories and classify all data instances, encompassing both known and novel classes. The authors approach CCD by substituting the prompt pool from L2P (Wang et al.) with a Gaussian Mixture Model (GMM). At every incremental step, both the last block of the backbone and the GMM module are updated. For prompting, the top-k Gaussian components (analogous to L2P) for each data instance are selected based on probability. The means of these components are then prepended to the input for inference. To mitigate the issue of forgetting, a subset of Gaussian samples is randomly chosen and forwarded to the next stage, allowing for dynamic expansion of the GMM modules. Additionally, the GMM module is coupled with a class number estimation method to gauge the count of emerging categories

**Strengths:**

- Expanding the prompt pool dynamically is beneficial and intuitively sound in a continual setting.
- Based on the experimental results, modifying the prompt pool using GMM appears promising. However, some confusion still needs to be addressed.
- Extensive experiments and comparison with SOTA methods.

**Weaknesses:**

- In the proposed method, the top-k "means" are treated as prompts. However, how should one interpret this? I am unclear about how adding the mean of the GMM can enhance the guidance provided to the backbone. It's worth noting that the GMM is detached from the backbone. Consequently, learning in the GMM module is somewhat disconnected from the optimization process. This is akin to integrating additional information, such as features, into the system.

- Much emphasis has been placed on the assertion that previous prompt methods are unsuitable for CCD because they require full supervision. The proposed method aims to address this by fitting the GMM. However, this claim might be overly assertive. The relaxation in data labels primarily stems from the two loss functions (both supervised and unsupervised) presented in Eqs. 2 and 3. Such techniques are commonly used in GCD, as noted by Vaze et al. (2022).

- The authors claim that 'these prompts facilitate adaptation to emerging data, thus preventing catastrophic forgetting.' However, in subsequent sections, the issue of forgetting is addressed by drawing samples from previous stages and reusing them in the second stage. This approach is reminiscent of a replay mechanism.

- Concerning the estimation of the novel class number, Table 3 provides interesting insights. Apart from the fine-grained CUB200, the estimation remains relatively consistent, irrespective of the class number across different stages. Even though the estimation does not align perfectly with the ground truth, the final performance might actually surpass scenarios where the class number is known, as illustrated by the comparison between Tiny200 in Tables 2 and 3. Has the study provided any rationale or explanation for this observation?

- Why is the final block of the backbone updated? Prompt learning aims to use prompts to 'instruct' the foundational model to access rich information without modifying it, as seen in methods like L2P and VPT[1]. It's worth noting that updating the backbone is geared towards acquiring new 'knowledge'. Is the overall performance improvement primarily attributed to the model being tailored to a specific dataset, or is it due to the 'instruction' provided by prompt learning?

* The description in the GMP module is somewhat confusing:
  - If I've interpreted it correctly, both the letter 'C' in Section 2.2 and the letter 'K' in Section 3.2 represent the number of classes. If that's the case, consistency in notation would be appreciated. Are the top-k prompts selected from among these C/K Gaussian components?
  - The method for obtaining GMM samples (as represented by the dots in Fig. 4) for optimizing the GMM is unclear.
  - How many samples advance to subsequent stages?
  - The algorithm appears to have inherent randomness, both from the selection of Gaussian samples and the random selection of reusable samples for later stages.

 [1] Jia et al. “Visual Prompt Tuning”. ECCV 2022.

**Questions:**

Which query function is used? I assume the cls token of the backbone output is used as in L2P?

---

> ### Author Response · Authors · 2023-11-23
>
> ## Response to reviewer si3G, *part 1*
>
> We would like to thank reviewer si3G for their careful review, insightful feedback, and acknowledgement of our novel and intuitively sound contributions, and extensive experiments. Here we would like to clarify some concerns addressed by reviewer si3G:
>
> - ### Q1: *How GMM’s mean components guide CCD model.*
> The motivation of our proposed model is to design a prompt module that can guide or instruct a self-supervised vision foundation model for better CCD. Moreover, not only can it guide the model, but we also believe there is a need to design a prompt module that suits the open-world nature of CCD with unlabelled data that contains unseen categories over time. Thus, intuitively we choose the Gaussian mixture model (GMM) as the candidate for our prompt design. By using the GMM mean component, it acts as a class prototype (to accommodate the absence of label information) which guides/enforces the model about the criteria / to look at (information/features that the model should care about) when making the decision to discover a category. We further show in **Appendix I**, that our GMM coupled with self-supervised vision foundation models actually produce prompts that are discriminative within different classes. Moreover, we conduct an experiment in Table 14, where instead of taking the “top-k” mean components/prompts, we vary the prompts “random-k” to observe the effect of difference prompt relevancy towards overall CCD performance. We can see that varying the prompt hurts the model’s performance, especially for the "NEW" ACC i.e., novel categories.
>
> - ### Q2: *Limitation of L2P and DualPrompt for CCD task.*
> Thanks for the comments. Please refer to **General response - Clarification on L2P & DualPrompt vs Our GMP**.
>
> - ### Q3: *Clarification on how GMM samples are used to prevent catastrophic forgetting.*
> Indeed, our method can be viewed as a replay method, unlike existing methods, it does not need to store any samples, but the parameter-efficient GMM can be used to generate an infinite number of replay samples on the fly. Also,  we use our replay differently than the usual replay-based method in the continual learning literature. Here the replay data is generated randomly by our GMM in each stage for each component and will be used to fit the next GMM which eventually builds/transfers previous GMM components which eventually leads to the formation of component means a.k.a prompts into the current stage GMM. Thus, by prompting these “previous components” from the previous replay samples, our model can tackle catastrophic forgetting.
>
> - ### Q4: *Rationale explanation regarding Table 3, main paper results.*
> Thank you very much for this comment. We followed this suggestion to investigate the cause of this phenomenon. While investigating the phenomenon between the TinyImageNet results obtained from both **Table 2 and Table 3**, we discovered a technical flaw in Table 3’s experiments Specifically, an incorrect training configuration was mistakenly loaded for training when using the GCD’s category number estimator. We have loaded the correct training configuration to retrain all models and updated the results in Table 3 accordingly. Based on the comparison of our model with other models in the unknown K settings in Table 3, we can see that our model outperforms all other models on all datasets under all metrics, except for the “OLD” accuracy of the Imagenet100 dataset. By comparing our proposed method in both Tables 2 and 3, we can observe that our proposed model has a slight decrease in accuracy for the “OLD” class when applying the estimated class number, while the “NEW” accuracy decreases more. This could be due to the prior introduced by the SS-Kmeans, which favours the “OLD” categories more. In addition, we conducted another experiment in **Appendix D, Table 7**, where we seamlessly adopted the GMM-based dynamic category number estimator into our CCD framework. We would like to emphasize that Table 7 aligns better with our proposed method and is therefore more preferred. As a result, we will swap Table 3 and Table 7 in our final manuscript.

---

> ### Author Response · Authors · 2023-11-23
>
> ## Response to reviewer si3G, *part 2*
>
> - ### Q5: *Clarification on why the final block of the backbone is updated.*
> Our model $H_{\theta}: \{\phi, f_{\theta}\}$ i.e., the self-supervised vision foundation model (in this case DINO) consists of $\phi$, an MLP projection head, and $f_{\theta}=\{f_e, f_b\}$ a transformer-based feature backbone that includes an input embedding layer $f_e$ and self-attention blocks $f_b$. During training, we only optimize $\phi$ and the final block of $f_b$. After training, we only require $f_{\theta}$ to extract the token classification i.e., $z\texttt{[CLS]}$ feature is used for clustering. If we fully freeze the backbone $f_b$, then unlike the L2P supervised continual learning implementation which has a final learnable fully connected layer for classification, our model will eventually not learn anything and act as a frozen DINO. Therefore, we further optimize the final block of the backbone to prevent this issue. For reference, we provide comparison results between frozen DINO, frozen DINO with learned L2P prompt pool and our proposed model in **Appendix H**. We can see that Our fine-tuned model outperforms the frozen models in all ACC metrics and for all datasets. Moreover, our fine-tuned model also appears to generalize better to data distributions (C100 & CUB200 datasets) that the DINO foundation model has not encountered before, which further justifies the design choice of our method for CCD.
>
> - ### Q6: *Clarifications of our GMP module.*
> 1. We thank the reviewer for their careful review,  the number of components in GMM is similar to the number of categories i.e, $C$ and the top-K prompts are selected among these $C$ Gaussian components.
> 2. To optimize GMM, we make use of classification token $z\texttt{[CLS]}$ extracted in the “gradient detach” mode of our own backbone $f_{\theta}$ as GMM samples.
> 3. Based on the ablation study the optimal number of samples is 100 samples per component.
> 4. For clarity we further provide a pseudo code of PromptCCD w/GMP model training in **Appendix A**.
>
> - ### Q7: *Clarification on query functions used on our proposed model.*
> As explained in **R-si3G-Q6 Clarifications of our GMP module sub-section (2)** we make use of our own backbone model $f_{\theta}$ as the query function to extract the classification token $z\texttt{[CLS]}$.

---

### Official Review · Reviewer_YA1L · 2023-11-01

**Soundness:** 2 fair
**Presentation:** 2 fair
**Contribution:** 2 fair
**Rating:** 5
**Confidence:** 4

**Summary:**

This paper proposes a prompt-based method for continual category discovery, which continually learns the keys of prompt by a GMM in an unsupervised manner. It also introduces a category estimation strategy based on GM when the number of new categories is unknown. The proposed method is evaluated on the benchmarks of continual GCD with comparisons to other baseline methods.

**Strengths:**

- The paper proposes a GMM-based replaying strategy for the prompt-based method to address the continual GCD problem, which is a new and reasonable design for the task.

- The experimental evaluation are conducted on both the settings of known and unknown class number with good results.

**Weaknesses:**

- The novelty of the paper is mainly on the proposed Gaussian Mixture Prompt Pool, which is rather limited. The authors argue that "the fixed-size prompt module restricts the model's ability to select prompts to a maximum of $M$",   but the proposed Gaussian Mixture Prompt Pool seems also a fixed-size prompt. In addition, what is the advantage of GMM compared to K-means, if we also store mean and covariance for replaying?

- The adaptation of L2P and Dual Prompt is not clear, which is vitally important since the main novelty is the design of keys. Specifically, how to learn the key of those two methods?

- In the prompt-based continual learning diagram, they only learn the prompt pool and fix the whole backbone. But this paper argues that "During training, only the final block of the vision transformer is finetuned .....".  What are the results if the backbone is fixed? Is the input query also learned by finetuning the vision transformer?

- The conclusion in ablation studies is incoherent. When the prompt selection is set to 5, and GMM sampling is set to 100, the "Spherical" outperforms "Diagonal". But the optimal number of prompts seems to be 10. The conclusion of ablation studies should be adjusted.

- Typo: there are many typo errors and some notation is unclear. For example, "Fourth, We hypothesize that categories may share commonalities regarding colour, shape, and other factors. ",  and $B$ and $B^L$ in Eqn(4). Please carefully check the manuscript.

**Questions:**

See the comments in the above.

---

> ### Author Response · Authors · 2023-11-23
>
> ## Response to reviewer YA1L
>
> We would like to thank reviewer YA1L for their careful review, insightful feedback,  acknowledgement of our model’s novelty and model promising results compared to other baseline models. Here we would like to clarify some concerns addressed by reviewer YA1L:
>
> - ### Q1: *Clarification of our GMP module.*
>
> Our Gaussian mixture prompt pool (GMP) differs from L2P and DualPrompt prompt pool in three ways. First, GMP parameters are not fixed as they depend on the number of C components (as shown in **Appendix H**), which is crucial in CCD tasks where the number of classes in the unlabelled data can grow or reduce over time. Second, we argue that GMM is advantageous over K-means when working with high/complex dimensional data. By using GMM, we allow a sample to have multiple memberships towards different cluster components, i.e., non-convex clusters. K-means assumes that the probability a sample belongs to a cluster is one, which in real-world data might not be true. Third, we show in **Appendix I** that GMM outperforms K-means for all clustering metrics, which means better prompt quality. Therefore, it is natural to choose GMM as a pool of prompts. For further discussion, please refer to **General response - Clarification on L2P & DualPrompt vs Our GMP**.
>
> - ### Q2: *Adaptation of L2P and DualPrompt for CCD.*
>
> To adapt L2P and DualPrompt as our baseline models for CCD as shown in **Section 2.1, main paper**, we make the following key modifications: (1) we change the pretrained model from a strong supervised pretrained model to DINO self-supervised pretrained model, (2) Originally in supervised continual settings, L2P and DualPrompt adopt supervised objective loss i.e., cross-entropy loss to optimize their models. For CCD, we modify the loss function into a contrastive learning objective, **Section 2.3, the main paper** combined with surrogate loss, **Eq.(1)** to optimize the prompt pool to pull selected keys close to corresponding query features. Please note that for our proposed method i.e., PromptCCD w/GMP, we do not use any surrogate loss as GMM is optimized separately. For details on how we optimize GMM, see **Appendix A**.
>
> - ### Q3: *Model optimization when the DINO backbone is fully frozen.*
>
> In the L2P model, the prompts are learnable parameters which are optimized jointly with the model. Unlike L2P, our prompts are not learnable as it is fitted from the features $z\texttt{[CLS]}$ which is learned by fine-tuning the final block of the backbone. Thus, if we fixed the backbone, then the only learnable parameter left in our model is the projection head which is not used during inference i.e. the model becomes fully frozen. Here we provide the comparison results between the frozen DINO w/GMP,  frozen DINO w/L2P and our proposed model in **Appendix H, Table 13**. We can see that our fine-tuned model substantially outperforms the frozen models. Moreover, the frozen models failed to generalize to distribution that DINO had not encountered before (i.e., CIFAR100 and CUB200, as DINO was pre-trained on ImageNet 1K). This led to a significant loss in accuracy compared to the fine-tuned models with average difference losses of 26.43% and 18.29% respectively. This observation further emphasizes the importance of our design choice.
>
> - ### Q4: *Clarification on model’s ablation studies.*
>
> Our model’s GMP ablation studies consider three factors: covariance type for GMM, number of prompts, and number of GMM sampling for replay. **Table 4** shows that the optimal GMM covariance types for CIFAR100 and CUB200 datasets are “Spherical” and “Full,” respectively, with five prompts and 100 GMM samples. Although there is some ambiguity on the covariance type, we set the default configurations to be “Diagonal” for the GMM covariance type, five prompts, and 100 GMM samples, which appears to be a good trade-off. We further perform an additional ablation study in **Appendix K** where we keep all covariance of GMMs to be spherical. The results show that 5 prompts and 100 samples lead to the best “ALL” accuracy

---

### Official Review · Reviewer_1nUF · 2023-11-01

**Soundness:** 3 good
**Presentation:** 2 fair
**Contribution:** 3 good
**Rating:** 6
**Confidence:** 3

**Summary:**

The paper presents a novel approach to the problem of continual category discovery (CCD), where the goal is to label objects in an unlabeled data stream that arrives over time, containing both known and new categories. The authors introduce PromptCCD, which incorporates a Gaussian Mixture Prompt Module (GMP) to dynamically update and guide data representation, mitigating forgetting. PromptCCD also features a Gaussian Mixture-based module for estimating categories within the unlabeled data, eliminating the need for prior knowledge of the number of categories. Additionally, the paper adapts the evaluation metrics from generalized category discovery for CCD and conducts comprehensive benchmarks.

**Strengths:**

1-It features an innovative Gaussian mixture prompt module that alleviates the need for label information and effectively addresses the issue of catastrophic forgetting.

2-The methodology liberates the model from depending on a predetermined number of categories, enhancing its adaptability to real-world scenarios.

3-The method is validated through extensive experimentation, where it consistently surpasses other state-of-the-art methods in performance.

4-The literature review is thorough, encompassing a wide array of existing works in the field, thus providing a solid foundation for the proposed approach.

**Weaknesses:**

1-The figures included are complex and lack clear, informative descriptions, necessitating repeated reference to the text for full comprehension, which disrupts the flow of understanding.

2-The method is described with an excessive level of detail that, while potentially beneficial, also burdens the reader with information overload. A more concise presentation, perhaps through structured pseudocode similar to that provided for the evaluation, could clarify the methodology more effectively. Additionally, the depiction of the method across Figures 2-4 is spread over multiple figures with captions that do not sufficiently convey the information, thereby diluting the explanatory power of the visuals.

**Questions:**

1-Given that a Gaussian Mixture Model (GMM) is employed, does the assumption of similarity between learning stages in conventional continual learning scenarios, which may vary greatly, potentially limit the model's capabilities?

2-Considering that a portion of the data is initially labeled, is it possible to leverage this labeled subset to learn a preliminary GMM that could subsequently be integrated into the broader GMM for the entire dataset?

---

> ### Author Response · Authors · 2023-11-23
>
> ##  Response to reviewer 1nUF
> We would like to thank reviewer 1nUF for their careful review, insightful feedback, and acknowledgement for recognising the innovation and novelty of our approach, comprehensive benchmark and extensive experiments, and comprehensive literature review. Here we would like to clarify some concerns addressed by reviewer 1nUF:
>
> - ### Q1: *Suggestion on paper presentations.*
>
> We thank reviewer 1nUF for their suggestions. We have revised the manuscript with updated  **Figures 2 & 3** with added captions and provide the pseudo-code to explain the overall process in **Appendix A**.
>
> - ### Q2: *Clarifications on how GMP is employed.*
>
> We used Gaussian mixture models (GMMs) even during the initial learning with labelled data. The GMM learned the features from the labelled data and transferred this knowledge in the form of GMM random samples and fit them to the next GMM.
>
> - ### Q3: *Discussion on the limitation of GMP.*
>
> As shown in **Appendix I, Table 13**, by learning GMM as a pool of prompt, and finetuned the final layer of the backbone, we observe the performance on C100 and CUB200 datasets to be interesting, we see that our proposed model guided by GMM’s mean components generalized better (compared to frozen DINO backbone) to datasets that the DINO foundation model has not encountered before which further justifies the design choice of our method. This argument is further justified in **Appendix H**, where we show that GMM’s prompts represent class prototypes. Thus, our method does not limit the model’s capability when the task identity for each stage is different.

---

### Official Review · Reviewer_k1JJ · 2023-11-02

**Soundness:** 3 good
**Presentation:** 3 good
**Contribution:** 3 good
**Rating:** 5
**Confidence:** 5

**Summary:**

This paper introduces a Gaussian Mixture Prompt Module (GMP) for Continual Class Discovery (CCD), a prompt learning technique that employs a Gaussian mixture model (GMM) as a dynamic prompt pool. The GMP overcomes the drawbacks of previous prompt learning methods by obviating the need for label information, adapting the parameterization based on data distribution, alleviating catastrophic forgetting, and exploiting shared commonalities among categories. The paper defines the optimization objectives for different learning stages, comprising supervised and unsupervised contrastive learning losses, as well as a surrogate loss to optimize the learnable prompt pool. The paper demonstrates that the proposed methods surpass existing approaches in terms of accuracy.

**Strengths:**

- The paper demonstrates the superiority of the proposed method over existing models by conducting a rigorous comparative analysis and achieving a substantial margin of improvement.
- The paper provides a comprehensive and lucid presentation of the technical details, including equations, tables, and descriptions of the methodology. The paper elucidates the underlying concepts and methods in an accessible manner, enabling readers to grasp the essence and rationale of the approach.
- The paper exhibits a high level of clarity and coherence in its organization and writing, facilitating the comprehension of the proposed approach and its merits.

**Weaknesses:**

- The method presented in this paper bears some resemblance to the one proposed in [1], but differs in that it tackles the CCD problem through incremental learning without any annotated information.
- This method employs DINO as the pre-training model. Although a similar pre-training approach was also adopted in [1], the problems addressed are different. The Continuous Learning task has annotation information, so the utilization of DINO’s self-supervised model is justified. However, in the CCD problem, there is no annotation information, and most incremental tasks are self-supervised. For models that are unsupervised, and the training data may have overlaps, using DINO is not equitable.
- The performance enhancement of this method stems more from DINO, and the paper lacks a comparison with methods without DINO pre-training models. The table in the paper does not specify the model's parameters used by different methods.

[1] Wang, Zifeng, et al. "Learning to prompt for continual learning." Proceedings of the IEEE/CVF Conference on Computer Vision and Pattern Recognition. 2022.

**Questions:**

- In Appendix C of the paper, the authors propose a method for dynamically adapting to the number of unknown categories, denoted as PromptCCD+. The method employed by PromptCCD in the main text to cope with unknown categories is to fix the number of categories to a predefined value.
- It would be beneficial to provide a direct comparison between the number of dynamically adapted unknown categories and the number of actual categories. This would offer a clearer insight into the effectiveness of the proposed method.
- Most of the experiments in the paper are conducted in three stages. Is it feasible to extend the method to more incremental stages?

---

> ### Author Response · Authors · 2023-11-23
>
> ## Response to reviewer k1JJ
> We would like to thank reviewer k1JJ for their careful review, insightful feedback, and acknowledgement of our rigorous analysis of the CCD task, high-level clarity and comprehensive presentation of our paper, and the effectiveness of our proposed method. Here we would like to clarify some concerns addressed by reviewer k1JJ:
>
>  - ### Q1: *Clarifications on the difference between the L2P method and our proposed method*
>
> In terms of the difference between the L2P method and our proposed method, please refer to **General response - Clarification on L2P & DualPrompt vs Our GMP**.
>
> - ### Q2: *The motivation for using the self-supervised vision foundation model for the CCD task*
>
> We would like to clarify that for L2P, they did not use a self-supervised pretrained model. Instead, they adopt a fully supervised pretrained model on very large-scale labelled data i.e., ImageNet-21K. In contrast, we use the fully self-supervised pretrained model DINO, which has been widely adopted in the static GCD task, by leveraging its strong feature generalization capability. DINO is a self-supervised (SSL) pretrained model (no labels are used during training). We want to point out that the SSL model has been widely adopted and justified in both NCD [**A**] and all GCD literature so far. It is a valid and reasonable choice to use such models because no supervision, such as class labels, was used to train the feature backbone.
>
> - ### Q3: *Information on the learnable parameters from all compared models.*
>
> We provide the table comparison of each model’s learnable parameters in **Appendix H**. Here we show that our model’s GMM parameters are flexible as we only need to define the number of components to scale it. Overall, compared to our baseline models (L2P and DualPrompt), Our proposed model’s learnable parameters are only 0.33% higher with category number C set to 100, which is still parameter efficient.
>
> - ### Q4: *Fixing the number of categories for main experiments.*
>
> Yes, in the main table experiments i.e., **Table 2**, we fix the number of categories to a predefined value for all models and datasets for fair comparison.
>
> - ### Q5: *Comparison between the predicted number of categories vs the actual number of categories.*
>
> We provide the tables for comparison between the predicted number of categories and the ground truth categories in **Table 3** (GCD [**B**] category number estimator) and **Appendix D** (Ours, by seamlessly adopting the GMM based dynamic category number estimator from GPC [**C**]). Overall, in this realistic scenario, our model still consistently outperforms other methods by a large margin across different datasets.
>
> - ### Q6: *The feasibility of extending the number of discovery stages.*
>
> For a fair comparison, we follow the previous work [**D**] data distribution i.e., 3 discovery stages. Also, we would like to clarify that our approach does not have any constraint on the number of discovery stages. In **Appendix E** we further compare our model with the recent ICCV-accepted papers called MetaGCD with 4 incremental category discovery stages and PAiGCD with 2 incremental category discovery stages.

---

### Author Response · Authors · 2023-11-23

# General response section, *part 1*
We sincerely thank all reviewers (R-k1JJ, R-1nUF, R-YA1L, R-si3G) for their insightful feedback, especially for recognizing the promising performance (k1JJ, 1nUF, YA1L, si3G), comprehensive experiments (k1JJ, 1nUF, si3G), clear motivation (k1JJ, si3G), and novelty of the proposed method (k1JJ, 1nUF, YA1L, si3G). Below, we list the changes we made to the paper and address a shared comment on clarification of our method vs L2P&DualPrompt. We address each individual reviewer’s comments separately. We will reflect more careful revision of the paper in the final version.


##  *1. Changes made in the revised manuscript*
We have made some changes towards our original submitted manuscripts. Here are the changes that we made:
1. We have revised **Figures 2 and 3** for better interpretability of the overall design of both baseline and proposed framework.
2. Revised explanation and wording for better clarity, removed typos and some repeated references addressed by the reviewer.
3. Added **Appendix A**: “Pseudo code for PromptCCD w/GMP”, to explain the procedures of PromptCCD w/GMP during model training.
4. Added **Appendix H**: “Why finetune the final block of DINO for CCD”, to justify why we need to optimize the final block of the DINO SSL pretrained model. It includes the information on learnable parameters for each compared model, **Table 12**; and the comparison between the fully frozen backbone models and the proposed finetuned models, **Table 13**.
5. Added **Appendix I**: “Ablation studies on prompt module design”, to analyze the effectiveness of GMP which acts as a pool of prompts, and the analysis of GMM clustering performance. It includes the comparison between learned GMM and K-means clustering performances given feature extractor from PromptCCD, GCD, and frozen DINO, **Figure 6**. We also show the confusion matrix on our PromptCCD w/GMP’s GMM clustering performance in both in **Figure 7** (coarse dataset) and **Figure 8** (fine-grained dataset).  Lastly, we study the effectiveness of taking “top-k” prompts rather than randomly taking prompts from GMM, **Table 14**.
6. Updated **Table 3** resolving training configuration’s flaws issue when implementing PromptCCD with GCD’s  “C” categories estimator.

---

> ### Author Response · Authors · 2023-11-23
>
> # General response section, *part 2*
>
> ## *2. Clarification on L2P & DualPrompt vs Our GMP (R-k1JJ, R-YA1L, R-si3G)*
> Our motivation is to design a prompt module that effectively guides the self-supervised vision foundation model in open-world continual category discovery. L2P [**E**] and DualPrompt [**F**] are two methods developed for supervised continual learning that utilizes a prompt pool to guide a strong supervised pretrained model during learning. Therefore, They are not applicable for the CCD task. However, as we show in **Section 2.1, the main paper** and **R-YA1L-Q2**, these models can be carefully modified for CCD with the necessary changes, despite being less effective than our method. However, there are a few critical limitations for these solutions. First, the representation learning only takes into account the unlabelled data in the current time step, which may lead to representation bias towards current data and disrupt the representation learned for the previous data. Second, the category discovery process is disjoint from the representation learning, therefore, lacking a proper mechanism for transferring knowledge from old classes to new classes, which is essential for the category discovery task. To address this issue, we proposed the novel GMM-inspired prompt called GMP (the details on the proposed method and the overall training process can be seen in **Section 2.2, main paper** and **Appendix A** respectively), Our proposed model introduces three significant improvements compared to existing models when tackling CCD (**R-k1JJ-Q1, R-YA1L-Q1, R-si3G-Q2**). Firstly, GMP’s prompt serves a dual role, namely (1) as task prompts to instruct the model (like in L2P and DualPrompt) and (2) as class prototypes  (as shown in **Appendix I**) to act as parametric replay sample distribution for discovered classes. The second role, which is unique and important for CCD/GCD, not only allows the model to draw unlimited replay samples to facilitate the representation tuning and class discovery in the next time step but also allows the model to transfer knowledge of previously discovered and novel categories and incorporate this information when making the decision to discover a novel category (**R-si3G-Q1**). Moreover, there are other limitations of L2P and DualPrompt when adapted to CCD. First, their fixed-size prompt modules can lead to parameter inefficiency and restrict the model’s ability to discover new categories and avoid forgetting. GMP can easily overcome this as the GMP module allows easy parameter adjustment and prevents forgetting by sampling and transferring samples to the next GMM at each time step. By transferring these random samples from the previous stage into the current GMM, we ensure that the learned components from the previous stage are retained and carried forward. This process of transferring the components effectively reduces the risk of forgetting previously learned information, enhancing the model's ability to preserve knowledge and adapt to new categories. Lastly, compared to L2P and DualPrompt, GMP is more adaptable to CCD tasks as it can be seamlessly combined with the  dynamic GMM-based category number estimator [**C**] without introducing any extra modules or parameters (See **Appendix D** for details and experiment results). Thus, by incorporating this dynamic category estimation without introducing any extra modules or parameters, our model becomes more adaptive and capable of handling the challenges posed by an ever-changing open world which allows our approach to effectively discover and accommodate new categories as they emerge, making it a valuable solution for real-world applications. Overall, our method is a highly optimized framework specialized in addressing the challenges in the CCD task, which can not be handled by L2P and DualPrompt.
>
> *Note that for supervised continual learning models such as L2P and DualPrompt adopt a strong supervised vision foundation model with incorporate label information during training.

---

> ### Author Response · Authors · 2023-11-23
>
> # General response section, *part 3*
>
> ## Reference:
>
> [A] Kai Han, Sylvestre-Alvise Rebuffi, Sebastien Ehrhardt, Andrea Vedaldi, and Andrew Zisserman. Autonovel: Automatically discovering and learning novel visual categories. IEEE TPAMI, 2021.
>
> [B] Sagar Vaze, Kai Han, Andrea Vedaldi, and Andrew Zisserman. Generalized category discovery. In CVPR, 2022.
>
> [C] Bingchen Zhao, Xin Wen, and Kai Han. Learning semi-supervised gaussian mixture models for generalized category discovery. In ICCV, 2023.
>
> [D] Xinwei Zhang, Jianwen Jiang, Yutong Feng, Zhi-Fan Wu, Xibin Zhao, Hai Wan, Mingqian Tang, Rong Jin, and Yue Gao. Grow and merge: A unified framework for continuous categories discovery. In NeurIPS, 2022.
>
> [E] Zifeng Wang, Zizhao Zhang, Chen-Yu Lee, Han Zhang, Ruoxi Sun, Xiaoqi Ren, Guolong Su, Vincent Perot, Jennifer Dy, and Tomas Pfister. Learning to prompt for continual learning. In CVPR, 2022b.
>
> [F] Zifeng Wang, Zizhao Zhang, Sayna Ebrahimi, Ruoxi Sun, Han Zhang, Chen-Yu Lee, Xiaoqi Ren, Guolong Su, Vincent Perot, Jennifer Dy, et al. Dualprompt: Complementary prompting for rehearsal-free continual learning. In ECCV, 2022a.

---

### Meta-Review · Area_Chair_uyqf · 2023-12-19

**Metareview:**

This paper studies the continual class discovery (CCD) problem, proposing a prompting-based method that utilizes a gaussian mixture model (GMM) to form a dynamic prompt pool. A series of supervised and self-supervised contrastive losses are used, in addition to a surrogate loss for the prompt pool. Results across a number of datasets are shown, as well as ablation studies.

While the reviewers appreciated the writing and experimental results demonstrated by the method, there were a number of significant concerns including the significance of the method compared to existing prompt-based works for continual learning and use of standard contrastive losses (k1JJ, YA1L, si3G), the relative significance of the method over just use of the better backbone (DINO) (k1JJ), and questionable results for example in terms of category estimation. While the rebuttal provided a number of responses to the questions, ultimately these most significant ones were not fully addressed. Crucially, some of the questionable results have led the authors to realize that there were errors in the configurations being run. This adds significant uncertainty as to the correctness of the overall paper since such debugging should not be done at the rebuttal stage.

 By the end, the scores are mixed at 5,6,5,3 with the most positive being the least confident. After having considered the paper, reviews, and rebuttals, I cannot recommend acceptance due to these significant issues, including lack of situation with respect to the space of works across CL and category discovery and especially the concerns raised about the ablations and experiments which have led to the discovery of errors. As a result, I encourage the authors to polish and thoroughly validate the results for a resubmission.

**Justification For Why Not Higher Score:**

Given the significant issues raised and especially the discovered errors, the authors should spend time validating the paper's results and findings, as well as situating it better, for resubmission.

**Justification For Why Not Lower Score:**

N/A

---

### Decision · Program_Chairs · 2024-01-16

Reject